# Crosstalk between regulatory elements in disordered TRPV4 N-terminus modulates lipid-dependent channel activity

Benedikt Goretzki[1,2], Christoph Wiedemann [1], Brett A. McCray [3], Stefan L. Schäfer [4], Jasmin Jansen[5,6], Frederike Tebbe[1], Sarah-Ana Mitrovic[7], Julia Nöth[7], Ainara Claveras Cabezudo [4,8], Jack K. Donohue[3], Cy M. Jeffries [9], Wieland Steinchen[10], Florian Stengel [5,6], Charlotte J. Sumner [3,11], Gerhard Hummer [4,12] & Ute A. Hellmich [1,2,13] ✉

Intrinsically disordered regions (IDRs) are essential for membrane receptor regulation but often remain unresolved in structural studies. TRPV4, a member of the TRP vanilloid channel family involved in thermo- and osmosensation, has a large N-terminal IDR of approximately 150 amino acids. With an integrated structural biology approach, we analyze the structural ensemble of the TRPV4 IDR and the network of antagonistic regulatory elements it encodes. These modulate channel activity in a hierarchical lipid-dependent manner through transient long-range interactions. A highly conserved autoinhibitory patch acts as a master regulator by competing with $PIP_2$ binding to attenuate channel activity. Molecular dynamics simulations show that loss of the interaction between the $PIP_2$-binding site and the membrane reduces the force exerted by the IDR on the structured core of TRPV4. This work demonstrates that IDR structural dynamics are coupled to TRPV4 activity and highlights the importance of IDRs for TRP channel function and regulation.

The majority of eukaryotic ion channels contain intrinsically disordered regions (IDRs), which play important roles in protein localization, channel function and the recruitment of regulatory interaction partners[1–3]. In some transient receptor potential (TRP) channels, IDRs make up more than half of the entire protein sequence[4]. Among the mammalian TRP vanilloid (TRPV) subfamily, TRPV4 has the largest N-terminal IDR, ranging from ~130 to ~150 amino acids in length depending on the species[4–6]. TRPV4 is a $Ca^{2+}$-permeable plasma membrane channel that is widely expressed in human tissues. It is remarkably promiscuous, and stimuli include pH, moderate heat, osmotic and mechanical stress, and various chemical compounds[7,8]. TRPV4 also garnered attention due to the large number of disease-causing mutations with distinct tissue-specific phenotypes primarily affecting the nervous and skeletal systems[9–13]. Among others, roles in cancer as well as viral and bacterial infections have also been described[14–16].

[1]Friedrich Schiller University Jena, Faculty of Chemistry and Earth Sciences, Institute of Organic Chemistry and Macromolecular Chemistry, Jena, Germany. [2]Centre for Biomolecular Magnetic Resonance (BMRZ), Goethe University, Frankfurt am Main, Germany. [3]Department of Neurology, Johns Hopkins University School of Medicine, Baltimore, MD, USA. [4]Department of Theoretical Biophysics, Max Planck Institute of Biophysics, Frankfurt am Main, Germany. [5]Department of Biology, University of Konstanz, Konstanz, Germany. [6]Konstanz Research School Chemical Biology, University of Konstanz, Konstanz, Germany. [7]Department of Chemistry, Section Biochemistry, Johannes Gutenberg-University Mainz, Mainz, Germany. [8]IMPRS on Cellular Biophysics, Frankfurt am Main, Germany. [9]European Molecular Biology Laboratory, EMBL Hamburg Unit, Deutsches Elektronen-Synchrotron, Hamburg, Germany. [10]Center for Synthetic Microbiology (SYNMIKRO) & Department of Chemistry, Philipps-University Marburg, Marburg, Germany. [11]Department of Neuroscience, Johns Hopkins University School of Medicine, Baltimore, MD, USA. [12]Institute of Biophysics, Goethe University Frankfurt, Frankfurt am Main, Germany. [13]Cluster of Excellence Balance of the Microverse, Friedrich Schiller University Jena, Jena, Germany. ✉e-mail: ute.hellmich@uni-jena.de

Crystal structures of the isolated TRPV4 ankyrin repeat domain (ARD) were among the first regions of a TRP channel to be resolved, showing a compact, globular protein domain with six ankyrin repeats[10,17,18]. Together with the IDR, the ARD forms the channel's cytoplasmic N-terminal domain (NTD). Furthermore, near full-length frog and human TRPV4 cryo-EM and X-ray crystallography structures are available, but lack the IDR, which was partially or fully deleted to facilitate structure determination or not visible in the final structure[19–22]. Using nuclear magnetic resonance (NMR) spectroscopy, we have previously shown that the TRPV4 IDR lacks appreciable secondary structure content[5]. Short stretches of N- and C-terminal IDRs were seen to interact with the ARD in TRPV channel cryo-EM structures[23–25], but no structural ensemble of a complete TRP(V) channel IDR has been visualized to date.

The TRPV4 NTD is responsible for channel sensitivity to changes in cell volume[26], its reaction to osmotic and mechanical stimuli[27,28] and the interaction with regulatory binding partners[29–32]. Therefore, a structural characterization of the TRPV4 NTD including its large IDR is critical to understanding TRPV4 regulation in detail.

To date, two regulatory elements in the N-terminal TRPV4 IDR have been described: (i) a proline-rich region directly preceding the ARD that enables protein-dependent channel desensitization;[29,30,32] and (ii) a phosphatidylinositol-4,5-bisphosphate (PIP$_2$)-binding site composed of a stretch of basic and aromatic residues directly N-terminal to the proline-rich region[33]. PIP$_2$ is a plasma membrane lipid and an important ion channel regulator[34,35]. In TRPV4, mutation of the PIP$_2$-binding site abrogates PIP$_2$-dependent channel sensitization in response to osmotic and thermal stimuli[33]. The lack of information on the role of additional IDR regions for channel regulation prompted us to investigate the TRPV4 IDR in more detail. In analogy to the integrative structural biology approach, which aims to build a consistent structure of a biological macromolecule or complex[36], the central aim of our study was to derive a cohesive functional model for TRPV4 regulation by its IDR through the integration of diverse experimental and computational methods on a range of TRPV4 deletion constructs and mutants.

In this work, we map hierarchically coupled regulatory elements linking the NTD's structural dynamics to channel activity along the entire length of the IDR and find them to modulate channel activity through lipid-dependent transient crosstalk. A highly conserved patch in the N-terminal half acts as an autoinhibitory element by modulating PIP$_2$ binding to the PIP$_2$-bindig site in the C-terminal half of the IDR. Using coarse-grained molecular dynamics (MD) simulations, we propose a force-dependent mechanism for how the conductive properties of the channel may be modulated via the ARD through a membrane-bound IDR. Furthermore, combining small angle X-ray scattering (SAXS), NMR and tryptophan fluorescence spectroscopy with crosslinking and H/D exchange mass spectrometry (XL-MS, HDX-MS) as well as atomistic MD simulations, we analyze the structural ensemble of the TRPV4 N-terminal domain and derive a structural model of a TRP channel with its extensive IDR. Our results highlight important regulatory functions of the IDR and underscore that the IDRs cannot be neglected when trying to understand TRP channel structure and function.

## Results

### Structural ensemble of the TRPV4 N-terminal intrinsically disordered region

To address the current lack of structural and dynamic information for the TRPV4 NTD, we purified the 382 amino acid *Gallus gallus* domain (residues 2–382, with 83/90% sequence identity/similarity to human TRPV4) as well as its isolated IDR (residues 2–134), and ARD (residues 135–382) (Fig. 1a–c, Supplementary Fig. 1). The avian proteins were chosen due to their increased stability compared to their human counterparts[29]. Analytical size-exclusion chromatography (SEC) and SEC-MALS (SEC multi-angle light scattering) showed that these

constructs are monomeric, while circular dichroism (CD) spectroscopy and the narrow chemical shift dispersion of the [$^1$H, $^{15}$N]-TROSY-HSQC NMR spectra of the $^{15}$N-labeled TRPV4 IDR in isolation or in the context of the NTD confirmed its high amount of disorder[5] (Fig. 1c–e; Supplementary Fig. 2). Chemical shift based secondary structure predictions showed that the TRPV4 IDR contains no appreciable secondary structure[5].

Small angle X-ray scattering (SAXS) probes the molecular shape of a molecule in solution. An IDR-containing protein is best described as a structural ensemble, rather than as a single structure, which can be analyzed by SEC-coupled small-angle X-ray scattering (SEC-SAXS) and subsequent Ensemble Optimization Method (EOM) analysis[37,38]. The isolated TRPV4 IDR is highly flexible and fluctuates between numerous conformations that, as a population, produce a skewed real-space scattering pair-distance distribution function, or $p(r)$ profile that extends to ~12.5–15 nm (Fig. 1f, Supplementary Fig. 3). Compared to a randomly generated pool of solvated, self-avoiding walk structures, the TRPV4 IDR preferentially sampled more compact states both in isolation and attached to the ARD, suggesting the presence of transient intradomain contacts (Fig. 1f–h, Supplementary Fig. 3). Interdomain contacts between the IDR and ARD were also apparent from the loss of signal intensities in the $^1$H, $^{15}$N-NMR spectra of the $^{15}$N-labeled IDR compared to the NTD (Supplementary Fig. 2b, c), e.g., for residues ~20–35 and ~55–115. The ARD itself was not resolved in the spectra of the NTD likely due to unfavorable dynamics.

### Structural dynamics of the TRPV4 ARD

The SAXS data, the dimensionless Kratky plot and the resulting $p(r)$ profile of the isolated ARD are typical of a more compact particle, especially in comparison to the IDR (Supplementary Fig. 3). Accordingly, the 28 kDa ARD has a significantly smaller radius of gyration ($R_g$ ~ 2.5 nm) and maximum particle dimension ($D_{max}$ ~ 11.5 nm) than the 15 kDa IDR ($R_g$ ~ 3.5 nm; $D_{max}$ ~ 12.5–15 nm). However, the SAXS data do not allow a straight-forward analysis of possible intradomain dynamics of the ARD.

To evaluate the structural flexibility of the ARD in solution, we used hydrogen/deuterium exchange mass spectrometry (HDX-MS) (Fig. 2a, Supplementary Table 3, Supplementary Data 1 and 2). HDX-MS probes the peptide bonds' amide proton exchange kinetics with the solvent and thus provides insights into the higher order structure of proteins and their conformational dynamics[39]. Both the isolated ARD and IDR domains and the entire TRPV4 NTD were subjected to HDX for different periods of time, and subsequently digested with porcine pepsin to assess where HDX occurred. The obtained peptides covered the entire ARD domain, but only a limited peptide coverage of the IDR could be achieved (i.e., 64.7%, see Supplementary Data 1 and 2) possibly owing to insufficient proteolytic digestion of the IDR or/and poor chromatographic properties or ionization of the generated peptides. For the IDR, rapid high HDX was apparent in the resolved parts (residues 3–24, 29–55, and 72–105) substantiating its unstructured character. The transient nature of interdomain contacts between ARD and IDR was underscored by the absence of a significant difference in the HDX of the individual domains in isolation or in the context of the full-length NTD.

Immediate high HDX was apparent for ARD loop 3 (residues 259–267), ankyrin repeat 5 (residues 319-327) and the linker between ankyrin repeats 5 and 6 (residues 344–348). Most α-helices showed progression in HDX over time except for α7 (repeat 3), α9/α10 (repeat 4), and α11/α12 (repeat 5) suggesting that these constitute the structural core of the ARD with the least flexibility. The peripheral ankyrin repeats 1, 2, and 6 underwent faster exchange (HDX at $10^3$ s). Complex conformational dynamics, i.e., slower motions in the ARD core and faster dynamics within the ARD loops and peripheral ankyrin repeats, are also in agreement with the extensive line broadening observed in the NTD's [$^1$H, $^{15}$N]-TROSY-HSQC NMR spectrum (Supplementary

Fig. 2b). Importantly, the HDX data for the ARD matches the dynamic pattern observed in our MD simulations (Fig. 2b, c) and indicate that the domain is stable in the sub-µs timescale, possibly an important prerequisite to transduce signals between IDR and transmembrane core. To investigate this further, we conducted atomistic MD simulations on the core *G. gallus* TRPV4 channel embedded in a lipid bilayer membrane (Fig. 2d, e, Supplementary Movie 1). Similar to its behavior in isolation, the ARD shows only subtle conformational fluctuations in the context of a TRPV4 homotetramer. Notably, protein dynamics in loop 3 and the C-terminal ankyrin repeat were reduced in the TRPV4 tetramer compared to the isolated ARD owing to their involvement in extensive intra- and inter-subunit contacts. Overall, our data suggest that the ARD is stable on the relevant time scales required to act as a putative signal transducer between IDR and transmembrane domain (see below).

## Long-range TRPV4 NTD interactions center on the PIP₂-binding site

Long-range interactions between IDR and ARD were investigated using crosslinking mass spectrometry (XL-MS). Except for the first 49 amino acids, 25 lysine residues are almost evenly distributed throughout the *G. gallus* TRPV4 NTD sequence. The lysine side chain amino groups can be crosslinked by disuccinimidyl suberate (DSS), probing $C_\alpha$-$C_\alpha$ distances up to 30 Å[40]. Both intradomain (within IDR or ARD) and interdomain (between IDR and ARD) crosslinks were observed for the NTD

(Fig. 3a, Supplementary Data 3). Many intra- and interdomain contacts were observed for the most N-terminal IDR lysine residues (K50, K56) and those within or close to the PIP₂-binding site on the C-terminal end of the IDR (K107, K116, K122). Importantly, these crosslinks were replicated in an equimolar mix of isolated IDR and ARD, supporting the involvement of these IDR regions in specific long-range interactions (Fig. 3b, c, Supplementary Data 3).

Conveniently, the TRPV4 PIP₂-binding site (consensus sequence KRWRR) important for TRPV4 sensitization[33] contains the sole tryptophan residue within the ~43 kDa NTD (W109 in our constructs).

Changes in the chemical environment of the PIP₂-binding site, e.g., through altered protein contacts, can thus be probed directly by differences in the tryptophan fluorescence spectra of deletion constructs generated around the PIP₂-binding site (Fig. 3d).

The fluorescence emission of a minimal construct (IDR^ΔN97, residues 97–134), which included the PIP₂-binding site and the proline-rich region, was suggestive of high solvent accessibility of the tryptophan residue and resembled that of free tryptophan in buffer (Fig. 3e, f). In longer constructs containing the ARD, additional parts of the IDR, or both, the fluorescence emission was blue shifted, indicating that W109 was in a more buried, hydrophobic environment. This effect was most pronounced for the full-length NTD, while deletion of the N-terminal half of the IDR (NTD^ΔN54) yielded an intermediate emission wavelength between full-length NTD and NTD^ΔN97, a construct comprising only the ARD, proline-rich region, and PIP₂-binding site. This indicates that the

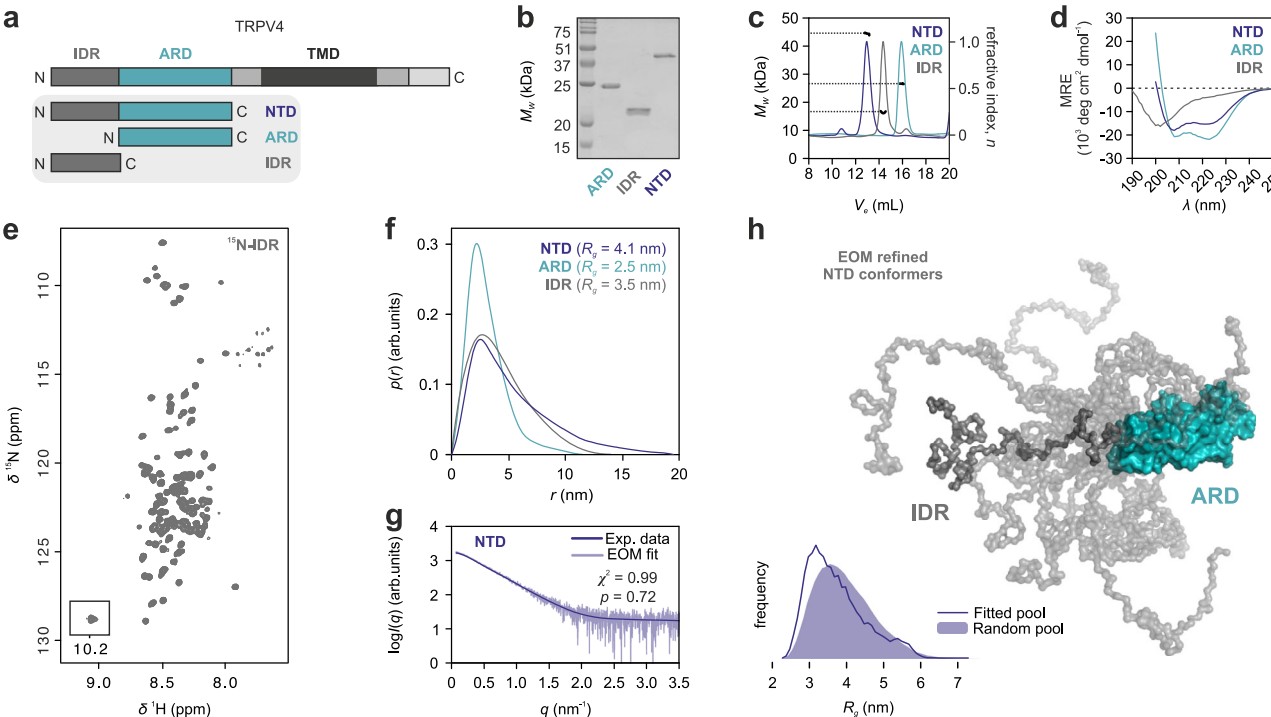

**Fig. 1 | Structural ensemble of the TRPV4 N-terminal domain. a** TRPV4 N-terminal constructs used for structural analyses. **b–d** Purified TRPV4 N-terminal constructs analyzed by Coomassie-stained SDS-PAGE **b**, SEC-MALS **c**, and CD spectroscopy **d**. SDS-PAGE in **b** comparing all constructs side by side was carried out once to evaluate sample purity and respective molecular weight. **e** [¹H, ¹⁵N]-TROSY-HSQC NMR spectrum of ¹⁵N-labeled TRPV4-IDR (see Supplementary Fig. 2 for backbone assignments). **f, g** SAXS pair-distance-distribution **f** and SAXS EOM (Ensemble Optimization Method) **g**, both in arbitrary units (arb. units), of TRPV4 N-terminal constructs (Supplementary Fig. 3). The real-space distance distribution yields a radius of gyration of $R_g = 3.4$ nm with a maximal particle dimension of $D_{max} = 14.0$ nm for the IDR, $R_g = 4.1$ nm and $D_{max} = 19$ nm for the NTD as well as $R_g = 2.5$ nm and a $D_{max} = 11.5$ nm for the ARD. Every protein exhibits levels of conformational heterogeneity and the $p(r)$ profiles should be interpreted as the

summed volume-fraction weighted contribution within the sample population, and not as single-particle distributions. The statistical analyses of the fit in **g** was carried out using the reduced $\chi^2$ method[93] (one-tailed distribution) and CorMap[64] (one-tail Schilling distribution) test methods. The determined $\chi^2$ and CorMap $p$ values are indicated in the corresponding graph. **h** NTD ensemble refined by EOM (Ensemble Optimization Method)[37,38]. Using a chain of dummy residues for the IDR and the X-ray structure of the TRPV4 ARD (PDB: 3W9G) as templates, a library of 10,000 NTD structures was generated and refined against the experimental data, allowing the comparison of the fitted versus the random pool and selecting a sub-set of ensemble-states representing the experimental data. Ten IDR conformers best representing the experimental scattering profile are depicted. Source data are provided as a Source Data file.

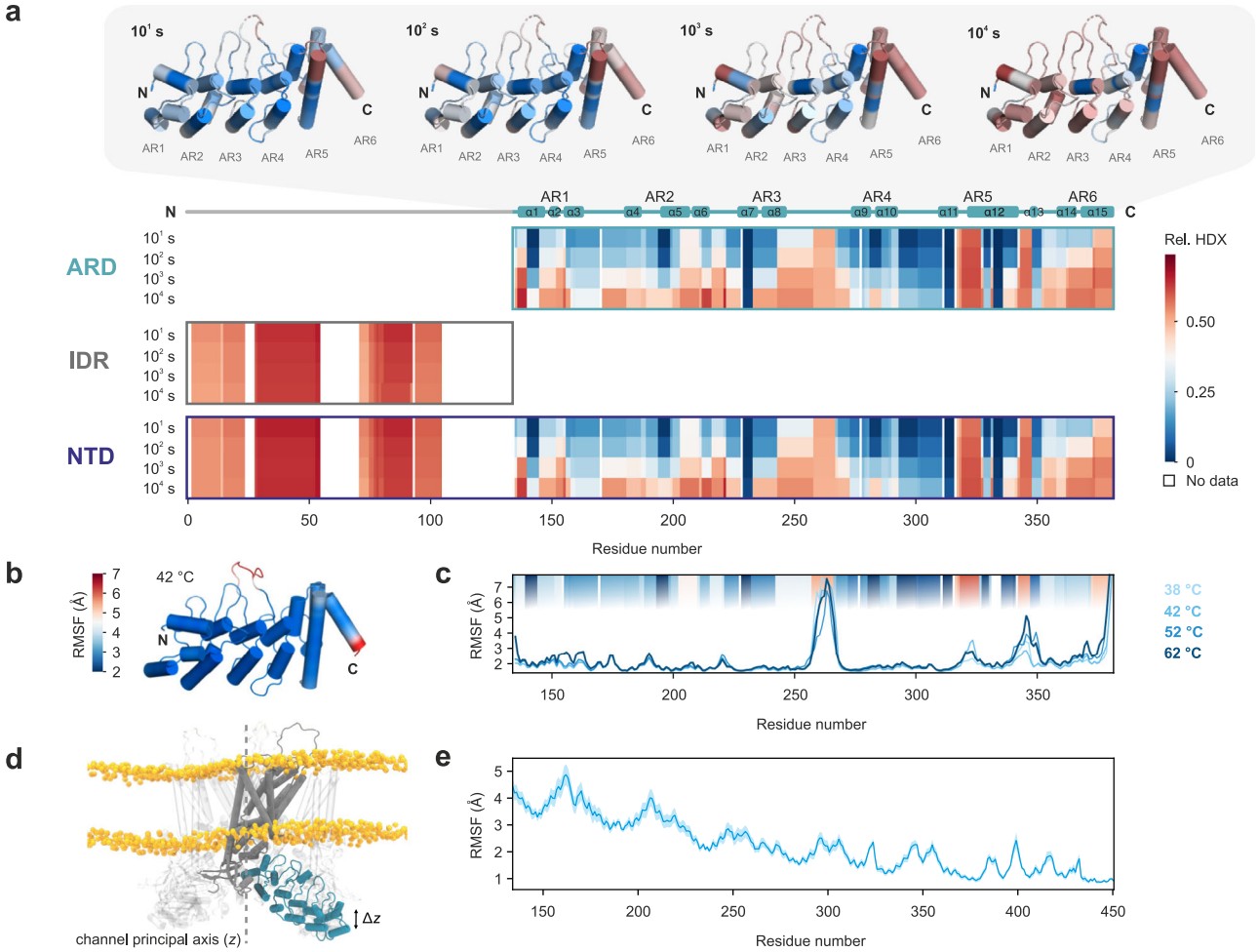

**Fig. 2 | Structural dynamics of the TRPV4 ARD. a** HDX of TRPV4 NTD and its isolated subdomains. Low (blue) to high (red) HDX shown for four time points. Areas without HDX assignment are colored white. For the ARD, HDX was visualized on the available X-ray structure of the *G. gallus* TRPV4 ARD (PDB: 3W9G). The six ankyrin repeats (AR) are indicated on top of the heat map diagram. **b, c** Root-mean-square fluctuations (RMSF) obtained from atomistic molecular dynamics (MD) simulations of the isolated *G. gallus* TRPV4 ARD in solution. RMSF at 42 °C mapped onto the ARD X-ray structure (PDB: 3W9G) **b** and RMSF per residue in simulations at increasing temperatures **c**. For comparison, HDX profiles after $10^2$ s from **a** are displayed in the plot background. **d, e** RMSF of the ARD with respect to the central TRPV4 axis obtained from 1 μs long MD simulations of the complete TRPV4 core (see also Supplementary Movie 1). **d** Schematic depiction of the MD simulation setup. The channel principal axis (defined as *z*) is indicated as a dashed vertical line. The RMSF was calculated as the square root of the variance of the motion along this axis (Δ*z*). **e** RMSF of TRPV4 residues 134–450 comprising the ARD. The solid line represents the average RMSF from all four protomers, the light area indicates the standard error of the mean. Source data are provided as a Source Data file.

local PIP$_2$-binding site environment is influenced by both the ARD and the distal IDR N-terminus.

## The PIP$_2$-binding site promotes compact NTD conformations

To probe the PIP$_2$-binding site's role for the NTD conformational ensemble, we replaced its basic residues by alanine (KRWRR → AAWAA) across TRPV4 N-terminal constructs (Supplementary Fig. 4a–d). CD spectroscopy and SEC showed that the structural integrity of the mutants was maintained (Supplementary Fig. 4c, d). However, we noticed consistently higher Stokes radii for the mutants compared to their native counterparts (Supplementary Fig. 4e), suggesting that the charge neutralization of the PIP$_2$-binding site affects the IDR structural ensemble. Interestingly, the $^1$H chemical shift of the W109 sidechain amide is different between the native IDR and IDR$^{\Delta N97}$. In contrast, this chemical shift is the same when comparing the respective PIP$_2$-binding site mutants (Fig. 3g, h). Thus, transient long-range interactions between the N-terminus and the PIP$_2$-binding site seem to be disrupted upon mutation of the PIP$_2$-binding site. Likewise, the tryptophan emission wavelength of the AAWAA mutants was also increased compared to the native constructs, indicative of a more solvent-exposed central tryptophan residue (Supplementary Fig. 4f, g).

NTD$^{AAWAA}$ and IDR$^{AAWAA}$ were also analyzed by SEC-SAXS and EOM (Fig. 4, Supplementary Fig. 4h–o). The scattering profile and real-space distribution of IDR$^{AAWAA}$ resembled the native IDR, indicating a random chain-like protein. However, the mutant's $R_g$ and $D_{max}$ values (3.5 nm and 14.5 nm, respectively) were slightly increased compared to the native IDR ($R_g$ = 3.4 nm and $D_{max}$ = 14.0 nm). This effect was even more pronounced in the context of the NTD, with $R_g$ and $D_{max}$ values of 4.5 nm and 19.5 nm, respectively, compared to $R_g$ = 4.1 nm and $D_{max}$ = 19.0 nm for the native IDR. Unlike the native IDR and NTD, the $R_g$ distributions of the mutant constructs obtained from EOM analysis agreed well with the randomly generated pools of solvated, self-avoiding walk structures (Fig. 4d). This suggests that constructs with a mutated PIP$_2$-binding site populate expanded conformations more frequently and show more random chain-like characteristics, thereby substantiating the role of the PIP$_2$-binding side as a central mediator of long-range contacts within the TRPV4 NTD.

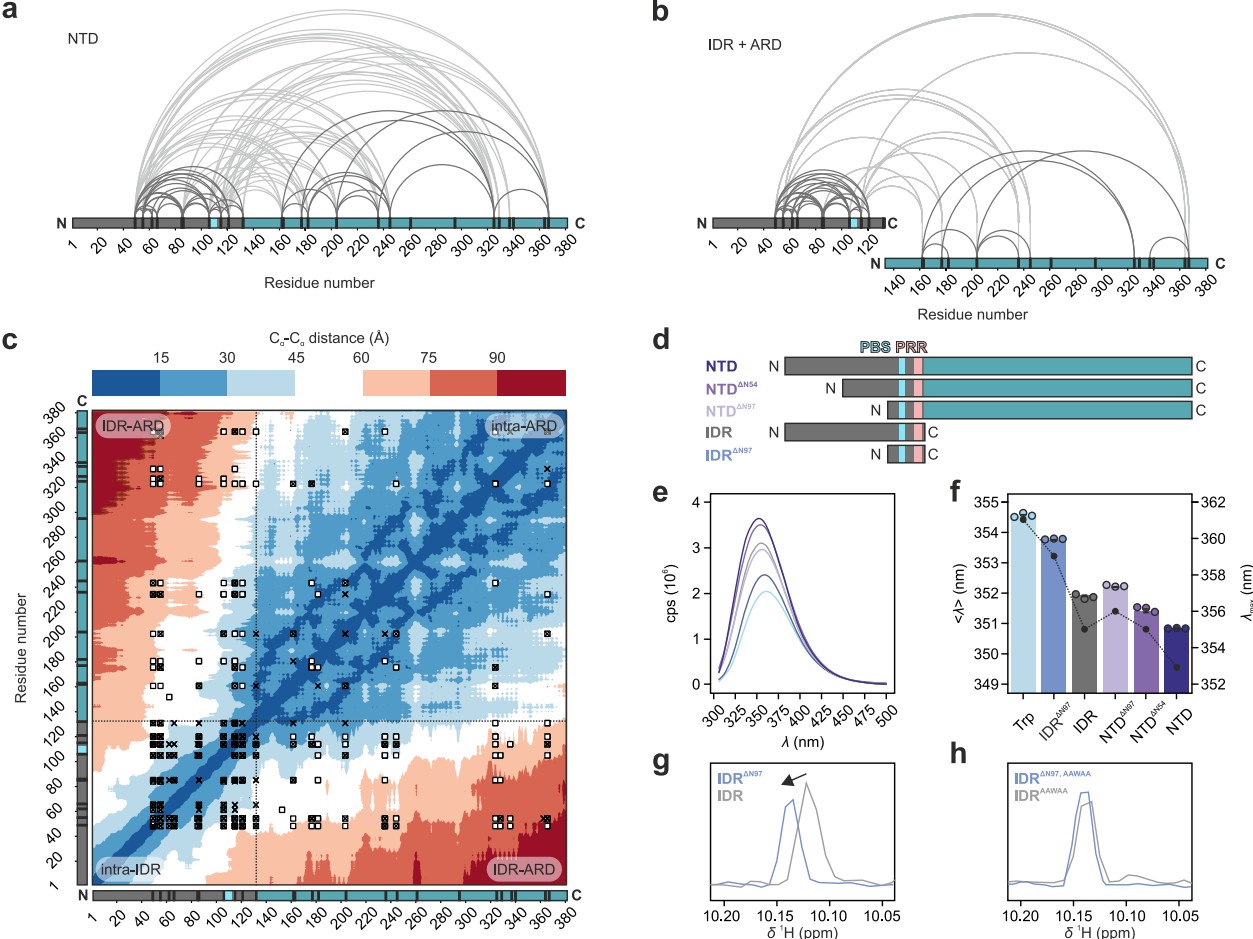

**Fig. 3 | Long-range intra- and interdomain interactions of the TRPV4 NTD.**
**a**, **b** Cross-linking mass spectrometry was used to probe interactions within and between IDR and ARD using either **a** the entire NTD or **b** isolated IDR (gray) and ARD (cyan) in a 1:1 ratio. Lysine residues are indicated by black tick marks, the PIP$_2$-binding site is marked light blue. Intradomain crosslinks are shown by curved lines in dark gray, interdomain crosslinks in light gray. **c** Heat map of C$_\alpha$-C$_\alpha$ distances for an NTD conformational ensemble consisting of 15 EOM-refined conformers based on SAXS data of the NTD (Fig. 1h). Crosslinks are highlighted by white squares (NTD), black crosses (equimolar ARD:IDR mixture) or white squares filled with black crosses (both experimental set-ups). **d** TRPV4 N-terminal constructs used for tryptophan fluorescence (PBS: PIP$_2$-binding site, PRR: proline rich region). **e**, **f** Tryptophan fluorescence spectroscopy of TRPV4 N-terminal constructs (IDR, NTD, NTD$^{\Delta N54}$ and NTD$^{\Delta N97}$ lacking the first 54 or 97 amino acids, respectively, and IDR$^{\Delta N97}$ (comprising PIP$_2$ binding site, surrounding basic residues and proline rich region) or isolated amino acid in buffer (Trp). Residue W109 in the PIP$_2$ binding site is the sole tryptophan residue in the entire NTD. Fluorescence intensity is presented in counts per second (cps). Bars represent the intensity weighted fluorescence emission wavelength <$\lambda$> (left axis). Data are presented as the mean value ± SEM from $n$ = 3 individual experiments. The fluorescence emission maximum $\lambda_{max}$ is shown by black circles connected through dotted lines (right axis). **g**, **h** $^1$H chemical shift differences of W109 sidechain amide between IDR and IDR$^{\Delta N97}$ as well as their respective counterparts harboring the PIP$_2$ binding site ($^{107}$KRWRR$^{111}$) mutation to $^{107}$AAWAA$^{111}$. Source data are provided as a Source Data file.

## Competing attractive and repulsive interactions between distinct IDR regions govern the NTD structural ensemble

The TRPV4 IDR consists of alternating highly conserved and non-conserved regions arranged along a charge gradient. An N-terminus rich in acidic residues segues into a C-terminus with an accumulation of basic residues followed by the proline-rich region connecting to the ARD (Fig. 5a, b). To probe the effects of differently charged and conserved IDR regions on the NTD structural ensemble, consecutive N-terminal deletion constructs were investigated by CD spectroscopy, SEC, and SEC-SAXS (Fig. 5c–e, Supplementary Figs. 5 and 6). The respective $R_S$, $R_g$ and $D_{max}$ values for consecutive N-terminal deletions do not change linearly, rather, depending on their charge ($z$), individual IDR regions mold the structural ensemble of the NTD differently (Fig. 5e). Addition of only the proline-rich region to the ARD (NTD$^{\Delta N120}$) notably increased the $R_S$, $R_g$ and $D_{max}$ values. This expansion is likely due to the formation of a polyproline helix[29]. A construct containing both the PIP$_2$-binding site and the basic residues preceding it (NTD$^{\Delta N97}$) showed an increase in compaction over the construct with the PIP$_2$-binding site alone (NTD$^{\Delta N104}$) suggesting cumulative effects of the regions surrounding the PIP$_2$-binding site for NTD structure compaction. Adding another ~40 residues yields NTD$^{\Delta N54}$, which includes the entire basic and highly conserved central stretch of the IDR, did not significantly increase the protein dimensions further underscoring the importance of the central IDR region for intra- and interdomain crosstalk. This is supported by NMR spectroscopy, where the region between residues 55 and 115 showed notable peak broadening in the context of the entire NTD compared to the isolated IDR (Supplementary Fig. 2c). Finally, the full-length NTD including the more acidic N-terminal IDR had significantly increased protein dimensions compared to NTD$^{\Delta N54}$, indicating that the overall structural ensemble of the TRPV4 NTD is modulated by competing attractive and repulsive influences exerted by distinct IDR regions.

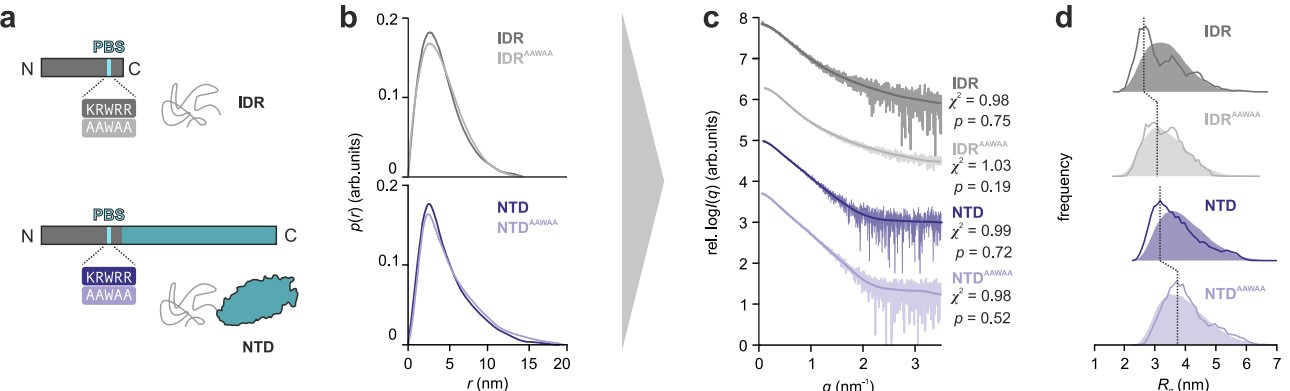

**Fig. 4 | The PIP$_2$-binding site promotes compact IDR conformations.**
**a** Constructs used in SEC-SAXS experiments. **b** Real-space pair-distance distribution functions, or $p(r)$ profiles, in arbitrary units (arb. units), calculated for IDR and IDR$^{AAWAA}$ (gray curves) as well as NTD and NTD$^{AAWAA}$ (blue curves). $p(r)$ functions were scaled to an area under the curve of 1. The real-space distance distribution of IDR$^{AAWAA}$ yields a radius of gyration of $R_g = 3.5$ nm with a maximal particle dimension of $D_{max} = 14.5$ nm (native IDR: $R_g = 3.4$ nm, $D_{max} = 14.0$ nm). NTD$^{AAWAA}$ has a $R_g = 4.5$ nm and a $D_{max} = 19.5$ nm (native NTD: $R_g = 4.1$ nm, $D_{max} = 19.0$ nm). **c** Fit

between EOM-refined IDR and NTD models and experimental scattering data, in arbitrary units (arb. units). The statistical analyses of the fits were carried out using the reduced $\chi^2$ method[93] (one-tailed distribution) and CorMap[64] (one-tail Schilling distribution) test methods. The determined $\chi^2$ and CorMap $p$ values are indicated in the corresponding graphs. **d** Comparison between $R_g$ values of IDR and NTD variants between random pool structure library (solid area) and EOM refined models (dotted line). Source data are provided as a Source Data file.

## A conserved patch in the N-terminal half of the IDR autoinhibits channel activity and acts on the PIP$_2$-binding site

To investigate the role of individual IDR regions on channel function, Ca$^{2+}$ imaging of human TRPV4 N-terminal deletion constructs expressed in the mouse motor neuron cell line MN-1 was performed as described previously[31] (Fig. 5f–i). All constructs were successfully targeted to the plasma membrane and structurally intact, as seen by the ability of the synthetic agonist 'GSK101'[41] to reliably activate the proteins (Fig. 5g, Supplementary Fig. 7a). All mutants had basal Ca$^{2+}$ levels similar to the full-length channel. Only *H. sapiens* TRPV4$^{\Delta N68}$ (corresponding to *G. gallus* TRPV4$^{\Delta N54}$) had strongly increased basal Ca$^{2+}$ levels (Fig. 5h). Deletion of the entire IDR (hsTRPV4$^{\Delta N148}$), as well as constructs retaining additional IDR regions, i.e., the proline-rich region (hsTRPV4$^{\Delta N133}$/ggTRPV4$^{\Delta N120}$), the PIP$_2$-binding site (hsTRPV4$^{\Delta N118}$/ ggTRPV4$^{\Delta N104}$) and the preceding basic residues (hsTRPV4$^{\Delta N111}$/ ggTRPV4$^{\Delta N97}$) yielded a channel non-excitable for osmotic stimuli (Fig. 5i). In contrast, hsTRPV4$^{\Delta N68}$ was hypersensitive to osmotic stimuli and its Ca$^{2+}$ influx far exceeded that of the native channel, indicating that the IDR N-terminus acts as a dominant autoinhibitory element. Furthermore, the data show that the PIP$_2$-binding site is not sufficient for osmotic channel activation but additionally requires the presence of the central IDR around residues ~68–111 (~54–97 in ggTRV4).

Nonetheless, we hypothesized that the site(s) in the N-terminal half of the IDR responsible for the observed channel autoinhibition and lipid binding attenuation may act via the channel's PIP$_2$-binding site. Thus, we compared the native IDR and IDR$^{AAWAA}$ NMR backbone amide chemical shifts to reveal interactions between the N- and C-terminal ends of the IDR (Fig. 6a). The largest chemical shift differences were naturally found in and around the mutated PIP$_2$-binding site itself and to a lower degree in the central IDR. Additionally, a region encompassing residues ~20–30 in the IDR N-terminus also showed notable chemical shift differences. This patch is the only conserved stretch in the N-terminal half of the IDR (consensus sequence FPLS-S/E-L-A/S-NLFE ($^{19/31}$FPLSSLANLFE$^{29/41}$ in gg/hsTRPV4)) (Fig. 6b, Supplementary Fig. 8).

Since our NMR data showed that the patch region is unstructured[5] (Supplementary Fig. 2), we replaced it with an (AG)$_5$ repeat to avoid α-helix formation (Supplementary Fig. 8). NMR relaxation data confirmed the absence of transient structure formation in the IDR$^{Patch}$ mutant (Supplementary Fig. 9). Importantly, the PIP$_2$-binding site residues' resonances in the IDR$^{Patch}$ mutant showed chemical shift changes compared to the native IDR (Fig. 6c), confirming that patch

and PIP$_2$-binding site on opposite ends of the IDR are in transient contact. The patch apparently also undergoes additional transient interactions with the ARD since the NMR signal intensities of residues in and around the patch region in the native IDR and IDR$^{AAWAA}$ constructs showed significant line broadening within the context of the NTD but not in the isolated IDR. This effect was abrogated in the IDR$^{Patch}$ mutant (Fig. 6d).

To elucidate the role of the patch for channel function, the full-length human TRPV4 channel harboring the patch mutant was expressed in MN-1 cells (Fig. 6e, f, Supplementary Figs. 7b and 8). Compared to the native channel, TRPV4$^{Patch}$ displayed significantly increased basal Ca$^{2+}$ levels and osmotic hyperexcitability, as previously seen for TRPV4$^{\Delta N68}$. This shows that the conserved patch in the N-terminal IDR is the dominant module responsible for autoinhibiting channel activity.

## The N-terminal half of the TRPV4 IDR modulates IDR lipid binding

Lipids and lipid-like molecules are important TRPV4 functional regulators[33,42–44], but beyond the PIP$_2$-binding site, lipid interactions with the TRPV4 NTD have not been probed in detail. A previously proposed lipid binding site in the ARD[42] seems implausible because it does not face the membrane in the context of the full-length channel[19,20]. Using the complementary N-terminal deletion mutants from our cell-based activity assay, we probed their interaction with liposomes (Fig. 7). Indeed, neither the isolated ARD, nor NTD$^{\Delta N120}$, also containing the proline-rich region, interacted with POPC/POPG liposomes in a sedimentation assay (Fig. 7a, b, Supplementary Fig. 10a, b). In contrast, ~75% of the native NTD was found bound to liposomes. For NTD$^{AAWAA}$, lipid binding was reduced to ~20%, indicating that the PIP$_2$-binding site is a major, but not the only lipid interaction site in the TRPV4 IDR. Deletion of the IDR N-terminal half slightly increased the fraction of lipid-bound protein. Incidentally, "protection" of the PIP$_2$-binding site from lipid binding by the N-terminal IDR was also observed with tryptophan fluorescence (Supplementary Fig. 10c–g) and may indicate that in the native IDR long-range intra-domain contacts compete with lipid binding.

NMR chemical shift perturbation assays allowed identification of the lipid-interacting IDR residues (Fig. 7c–e, see Supplementary Fig. 11 for $^{13}$C, $^{15}$N-labeled IDR$^{AAWAA}$ backbone assignments). In the native IDR, ~75% of all residues showed line-broadening in the presence of POPG-containing liposomes. Coarse-grained MD

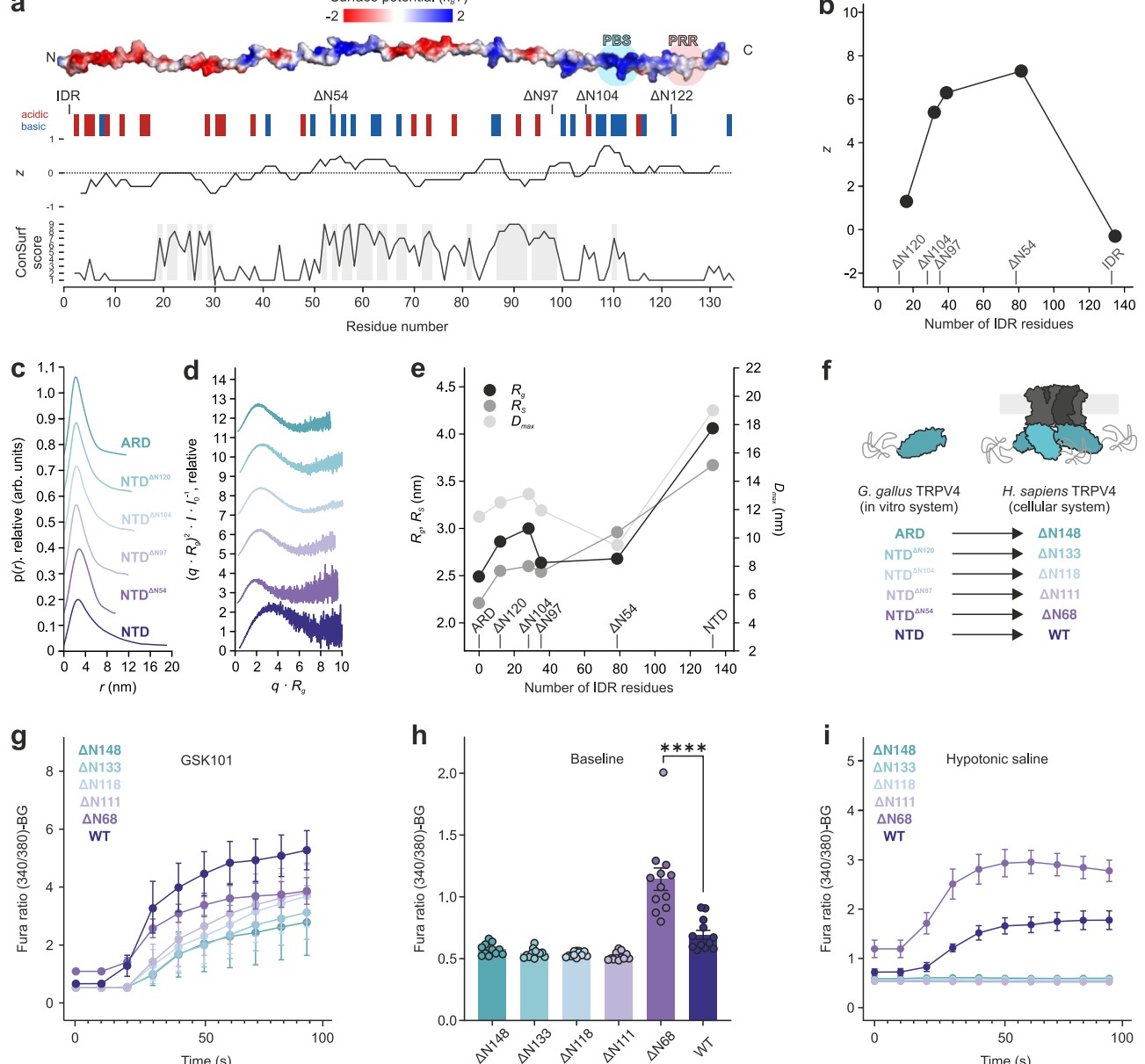

**Fig. 5 | The distal N-terminus affects the structural NTD ensemble and TRPV4 channel activity. a** Topology of NTD truncations showing the charge distribution $z$ (www.bioinformatics.nl/cgi-bin/emboss/charge) and sequence conservation (ConSurf[50]) along the IDR. **b** Overall charge ($z$) in the IDR at physiological pH (7.4) depending on the IDR length (values determined with ProtPi, www.protpi.ch). **c** Normalized real-space distance distribution $p(r)$, in arbitrary units (arb. units), of NTD and NTD deletion constructs. **d** Dimensionless Kratky plot of NTD and NTD deletion mutants. **e** Radius of gyration ($R_g$) and Stokes radius ($R_S$) determined from the real-space distance distribution in **a** and the SEC analysis (Supplementary Fig. 5c), respectively, plotted versus the number of IDR residues the NTD constructs. The maximum particle dimension ($D_{max}$) is plotted on the right y-axis. **f** N-terminal deletion mutants in the in vitro (*G. gallus*) and *in cellulo* (*H. sapiens*)

systems. **g** Activation of TRPV4 constructs with the synthetic agonist GSK101 shows plasma membrane targeting and structural integrity of all constructs. Data are presented as mean values ± SEM from $n = 6$ biologically independent experiments, each with 10–30 cells per field of view. **h** Basal $Ca^{2+}$ levels in MN-1 cells expressing different TRPV4 constructs. Data are presented as mean values ± SEM from $n = 12$ biologically independent experiments, each with 10–30 cells per field of view. The **** indicates a $p$ value of $p < 0.0001$ (one-way ANOVA with Dunnett's multiple comparison test). **i** Stimulation of $Ca^{2+}$ flux by hypotonic saline at $t = 20$ s in MN-1 cells expressing different TRPV4 constructs. Data are presented as mean values ± SEM from $n = 6$ biologically independent experiments, each with 10–30 cells per field of view. Source data are provided as a Source Data file.

simulations of the IDR on a plasma membrane mimetic corroborated the NMR experiments (Supplementary Fig. 12). Lipid interactions were seen to be dominated by the PIP₂ binding site, the central IDR and a conserved N-terminal patch (see below). In MD simulations with IDR^AAWAA, lipid interactions were severely reduced in the PIP₂-binding site, again agreeing with the NMR data (Fig. 7e, Supplementary Fig. 10, Supplementary Fig. 12).

Indicating an electrostatic contribution, an increase in salt concentration or the use of net-neutral POPC liposomes in NMR experiments reduced the observed line broadening for both native IDR and IDR^AAWAA (Fig. 7d, e, Supplementary Fig. 12c). Interestingly, the MD simulations also suggested a general preference for negatively charged PIP₂ over other membrane constituents for both the PIP₂-binding site and the central IDR (Supplementary Fig. 12).

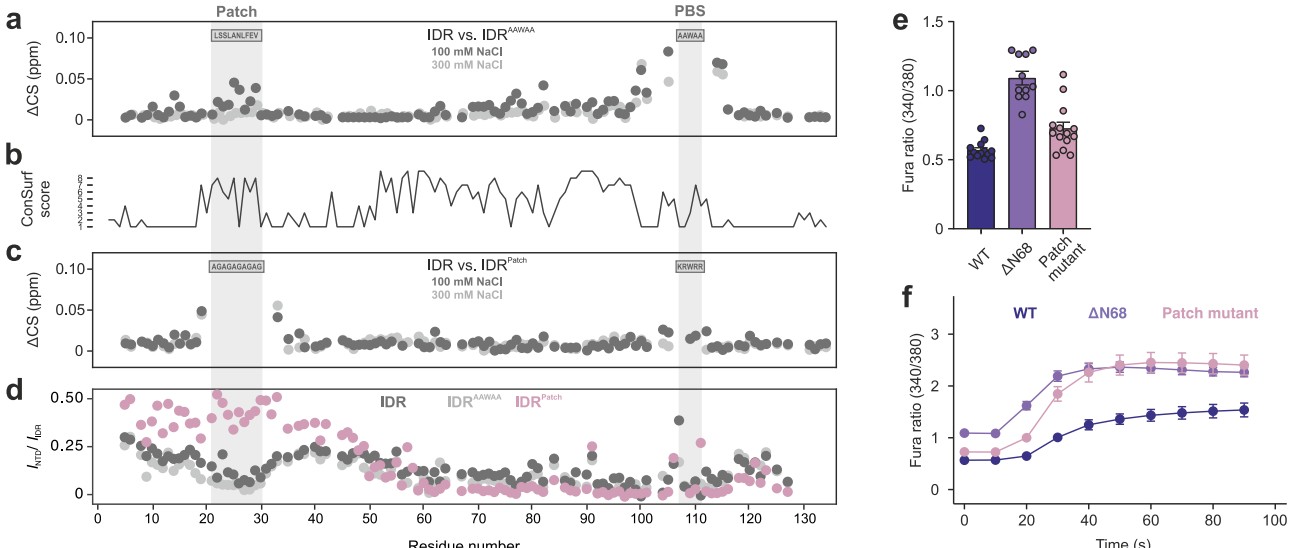

Fig. 6 | A highly conserved patch in the N-terminal TRPV4 IDR transiently interacts with the C-terminal PIP₂ binding site and autoinhibits TRPV4 function. a Chemical shift differences at high and low salt between $^{15}$N-labeled native IDR and IDR$^{AAWAA}$ with a mutated PIP₂ binding site (PBS) shows that mutagenesis of the PBS leads to chemical shift changes in the conserved N-terminal patch. At the higher salt concentration (light gray), these chemical shift perturbations are significantly reduced. b Degree of conservation in TRPV4 IDR determined with ConSurf[50] (Supplementary Fig. 11). c Chemical shift differences at high and low salt between $^{15}$N-labeled IDR and IDR$^{Patch}$ with a mutation in the conserved N-terminal patch shows that mutagenesis leads to chemical shift changes in the PBS. At the higher salt concentration (light gray), these chemical shift perturbations are significantly reduced. d Relative peak intensity of IDR, IDR$^{AAWAA}$ and IDR$^{Patch}$ residues in the isolated IDR or in context of the ARD (i.e., NTD, NTD$^{AAWAA}$ or NTD$^{Patch}$). All protein concentrations used were 100 μM. A value of 0.5 indicates that peak intensities for a respective IDR residue are halved when the ARD is present, a value of zero represents complete line broadening in the context of the NTD. Accordingly, lower values are indicative of IDR/ARD interactions. e, f Ca$^{2+}$ imaging of hsTRPV4 variants expressed in MN-1 cells. e Basal Ca$^{2+}$ and f hypotonic treatment at $t = 20$ s show increased activity of the patch mutant. For better comparison, data for TRPV4$^{ΔN68}$ are replotted from Fig. 5h, i. Data in e are presented as mean values ± SEM from $n = 13$ (TRPV4), 11 (TRPV4$^{ΔN68}$), 14 (TRPV4$^{Patch}$) and in f from $n = 12$ (TRPV4, TRPV4$^{ΔN68}$, and TRPV4$^{Patch}$) biologically independent experiments, each with 10–30 cells per field of view. Source data are provided as a Source Data file.

## The conserved patch competes with PIP₂ for binding to the PIP₂-binding site

To explore the extent of PIP₂ binding to the IDR, and the possible role of the autoinhibitory patch, we carried out NMR chemical shift perturbation assays with $^{15}$N-labeled IDR variants (Fig. 8a–c). In the native IDR, residues within and around the PIP₂-binding site (residues ~100–115), the central IDR (residues ~55–100) and the autoinhibitory patch (residues ~20–30) show the strongest responses to diC8-PIP₂ addition (Fig. 8a). The effect of PIP₂ on the autoinhibitory patch (Supplementary Fig. 8a) can be explained by the interactions of the patch with the PIP₂-binding site detected in the NMR experiments (Fig. 6a) and the coarse-grained MD simulations of the membrane-bound IDR (Fig. 8d, S12a–c). In the simulations, a substantial local increase in PIP₂ was observed around the PIP₂-binding site and the central IDR, but not in the patch region. This indicates that the observed NMR chemical shifts within the N-terminal patch are secondary effects, presumably based on altered protein-protein interactions upon PIP₂ addition. Notably, in the native IDR, both PIP₂-binding site and patch showed a similar dose response to PIP₂ as gauged by the similar degree of line broadening for these regions (Fig. 8a, gray bars). This suggests that PIP₂-binding site and patch act in concert and that lipid interactions in the PIP₂-binding site are also sensed by the autoinhibitory patch.

For IDR$^{AAWAA}$, dampened spectral responses to PIP₂ were observed in the mutated PIP₂-binding site, large parts of the central IDR and, to a much lesser degree, in the patch region (Fig. 8b) suggesting reduced coupling between these regions when the PIP₂-binding site is mutated. Likewise, in MD simulations of IDR$^{AAWAA}$, PIP₂ binding was largely abrogated in the mutated PIP₂-binding site (Fig. 8d, Supplementary Fig. 12c). Consequently, the mutated PIP₂-binding site frequently lost and regained contact with the lipid bilayer, although other IDR regions remained attached to the membrane throughout the simulations.

Mutation of the patch did not alter the lipid interaction pattern with the central IDR and PIP₂-binding site per se as gauged by NMR spectroscopy and MD simulations (Fig. 8c, d, Supplementary Figs. 8b, c, 9e, and 12c). Since the severe line broadening in the $^1$H, $^{15}$N-NMR IDR spectra precluded a more detailed analysis, we also took advantage of the PIP₂ headgroup phosphate groups as a $^{31}$P NMR reporter (Fig. 8e). In agreement with the liposome sedimentation assay (Fig. 7), both the deletion (IDR$^{ΔN97}$) or mutation of the patch (IDR$^{Patch}$) mutation increased PIP₂ binding compared to the native IDR as gauged by the extent of the respective $^{31}$P chemical shifts (Fig. 8f). Thus, the N-terminal patch seems to compete with PIP₂ lipids for the PIP₂-binding site via transient protein-protein interactions, thereby suppressing channel activity.

## Membrane-bound PIP₂-binding site exerts a pull force on the ARD

It remains unclear how lipid binding to the IDR is transduced to the structured core of TRPV4 to modulate the conductive properties of the channel. We speculated that IDR-lipid interactions could alter the channel gating properties by exerting a pull-force on the ARD. To probe this, we carried out coarse-grained MD simulations, in which the IDR's C-terminal residue (V134) was kept at distances of 5-9 nm below the membrane to emulate the distance constraints enforced on an ARD-anchored IDR in a full-length channel (Fig. 9a). From the mean restraint forces for native IDR, IDR$^{AAWAA}$ and IDR$^{Patch}$, we determined force-displacement curves as function of the height of V134 over the membrane center (Fig. 9b).

At heights <6.5 nm, the force on the C-terminal end of the IDR probes the membrane interaction of basic residues C-terminal of the PIP₂-binding site. Consequently, all constructs experienced similar forces. At a height of ~6.5 nm, the residues C-terminal of the PIP₂-binding site detached from the membrane in all IDR constructs. Thus,

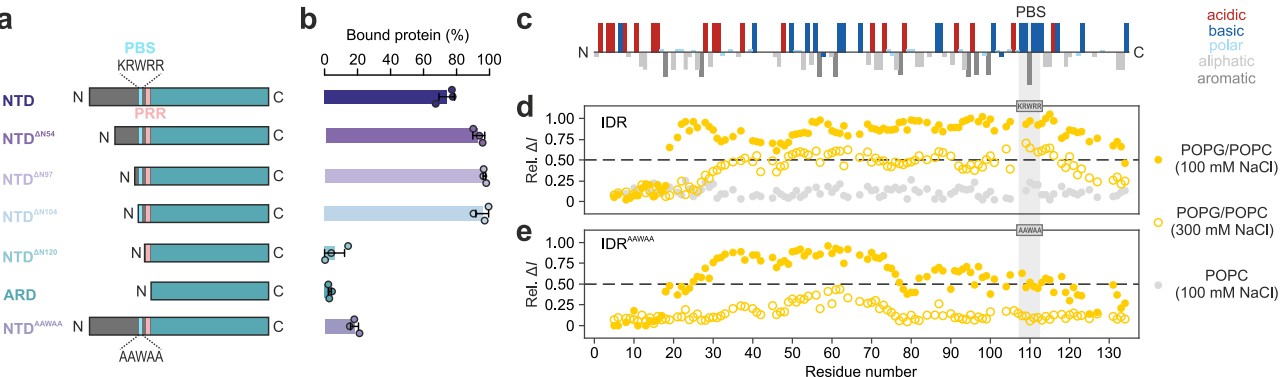

**Fig. 7 | Extensive lipid binding in the IDR is negatively affected by the distal N-terminus.** **a** Topology of N-terminal deletion mutants used for liposome sedimentation assay. **b** Protein distribution between pellet ("bound protein") or supernatant fraction after centrifugation, quantified via densitometry of SDS-PAGE protein bands using imageJ[51]. Data are presented as the mean value ± SEM from $n = 3$ individual experiments. **c** Distribution of charged and hydrophobic residues in the TRPV4-IDR shows a gradient of a consecutively more basic and hydrophobic protein from N- to C-terminus. Plotted with the PepCalc tool (https://pepcalc.com/). **d, e** NMR signal intensity differences for $^{15}$N-labeled IDR variants (100 μM) in the absence and presence of POPC (light gray circles) or POPC-POPG containing liposomes at low (filled yellow circles) or high salt concentration (open yellow circles). Higher values are indicative of lipid binding. Source data are provided as a Source Data file.

forces at heights ≥7 nm are generated by the PIP$_2$-binding site pulling at the membrane. Importantly, the PIP$_2$-binding site remained membrane-bound over the entire height regime in the simulations with native IDR and IDR$^{Patch}$. In contrast, these interactions were lost in the IDR$^{AAWAA}$ mutant, resulting in greatly reduced pull forces (~10 versus -17.5 pN at 9 nm) (Fig. 9b, c). Furthermore, the reduced slope of the near-linear force-height curve beyond 6.5 nm implies a four-fold higher effective force constant acting on the IDR C-terminus with an intact PIP$_2$-binding site compared to IDR$^{AAWAA}$. We also noticed a slightly lower slope and weaker forces for IDR$^{Patch}$ compared to the native IDR.

The forces observed here for an IDR C-terminus linked to a membrane-bound PIP$_2$-binding site are in the regime reported for other biochemical processes[45,46]. However, the smoothened energy landscape in our coarse-grained simulations may underestimate the actual force exerted on the ARD by its membrane-bound IDR "anchor". The strength of the PIP$_2$-binding site interaction with the membrane also became apparent when constraining the IDR C-terminus at heights >8 nm. Here, rather than detaching, the pull of the PIP$_2$-binding site led to noticeable membrane deformations (Supplementary Fig. 12a). TRPV4 may thus not only be able to sense, but under certain conditions also directly affect its membrane microenvironment via its IDR.

**The cytosolic TRPV4 N-terminal 'belt' more than doubles channel dimensions beyond the structured ion channel core**

Structural information for TRP channel IDRs is incomplete at best since they are not amenable to X-ray crystallography or cryo-electron microscopy studies due to their inherent spatiotemporal flexibility[4]. We previously calculated the dimensions theoretically sampled by TRPV channel IDRs assuming they behaved as unrestrained worm-like chains and found that fully expanded IDRs may contribute an additional 5–7 nm end-to-end distance to the structured cytosolic domains[4]. Here, we combined the results from SAXS, NMR and MD simulations to derive an experimentally-informed structural model of the TRPV4 ion channel. We found that the cytosolic "belt" formed by the TRPV4 N-terminal IDRs is smaller than previously anticipated due to the extensive lipid and intradomain interactions of the IDR. Nonetheless, the N-terminal IDRs more than double the TRPV4 diameter along the membrane plane from approximately 140 Å to a maximum of ~340 Å (Fig. 10a, b, Fig S13, Supplementary Movies 2 and 3). With their IDRs, these proteins thus dramatically extend their reach and may act as multivalent cellular recruitments hubs.

## Discussion

In this study, we show that the TRPV4 N-terminal IDR encodes a network of transiently coupled regulatory elements that engage in hierarchical long-range crosstalk and can enhance or suppress TRPV4 activity (Fig. 10c). Such contacts may affect lipid binding as seen here, but presumably can also be modulated by other ligands[47], regulatory proteins[29–32,47] or post-translational modifications[48] within the ARD and IDR to enable a fine-tuned integration of multi-parameter inputs by TRPV4.

While the PIP$_2$-binding site is crucial for channel activity[33,44], we observed that N-terminal IDR deletion mutants retaining the PIP$_2$-binding are still inactive if they do not also include the central IDR (Fig. 5i). Accordingly, this region may function in two ways: by enriching PIP$_2$ in the channel vicinity and by increasing the IDR's residency time at the plasma membrane. Furthermore, we discovered an autoinhibitory patch that suppresses channel activity by acting on the PIP$_2$-binding site.

Structural studies on TRPV channels have found that the ARDs can undergo concerted motions in response to ligand binding that may be connected to channel activity[24], with an unclear role of the IDR. Lipid interactions in the IDR are important regulators of TRPV4 activity[33,44], and here we find them to depend on intradomain interactions within the TRPV4 N-terminus (Figs. 7 and 8). Our MD simulations suggest that lipid-binding of the IDR at the PIP$_2$-binding site exerts a pull-force on the ARD, which itself is sufficiently rigid to further transduce mechanical forces to the TRPV4 core. The ARD is connected to the PIP$_2$-binding site via a proline-rich region which forms a poly-proline helix[29]. The proline-rich region may thus be a relatively stiff connector to efficiently transduce pull forces between ARD and membrane-bound PIP$_2$-binding site as suggested by our MD simulations (Fig. 9). Our NMR experiments show that the proline-rich region is not affected by lipids itself (Figs. 7d and 8a). However, it binds the channel desensitizer PACSIN3[29,30,32], which may affect the interaction between membrane-bound IDR and the structured channel core. Likewise, the N-terminal autoinhibitory patch may reduce the pull force exerted by the PIP$_2$-binding site by competing with its ability to bind lipids and thus effectively dampen channel activity. Our data thus provide a mechanistic explanation for prior observations that TRPV4 variants lacking part of the distal N-terminus display osmotic hypersensitivity[28,49]. Furthermore, a conformational equilibrium between PIP$_2$-binding site interaction between membrane and autoinhibitory patch may allow TRPV4 to fine-tune channel responses depending on cell state and regulatory partners (Fig. 10d).

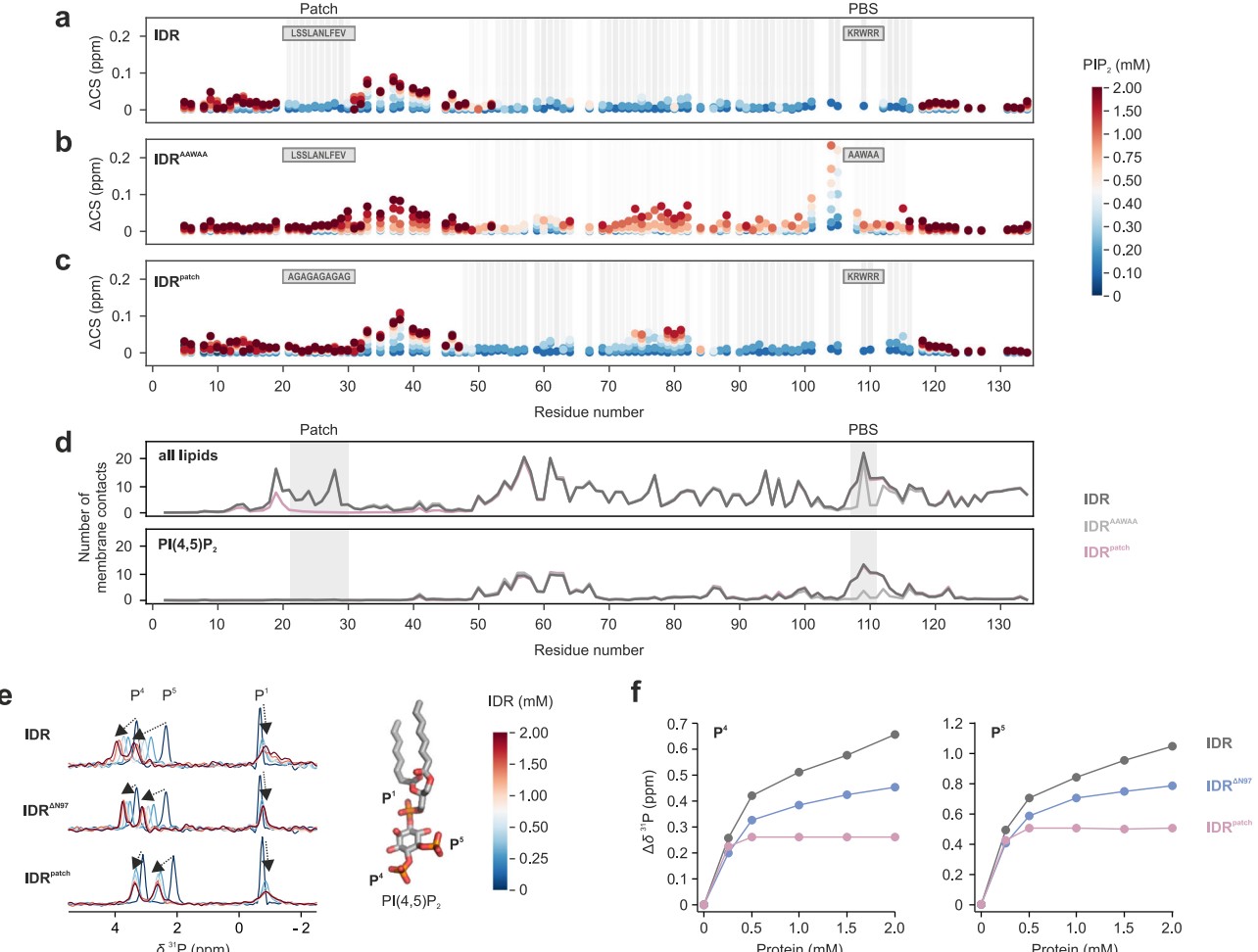

**Fig. 8 | The conserved N-terminal patch modulates lipid binding to the IDR.** **a**, **b**, **c** Chemical shift perturbation of $^{15}$N-labeled **a** IDR, **b** IDR$^{AAWAA}$, and **c** IDR$^{Patch}$ titrated with short-chain PIP$_2$. Mutated regions are indicated in gray boxes, chemical shift changes are depicted by colored spheres, residues showing line broadening are highlighted by gray bars. **d** Average number of membrane contacts for each residue of the native IDR (dark gray), the IDR$^{AAWAA}$ mutant (light gray) and the IDR$^{Patch}$ mutant (mauve) on a lipid bilayer composed of POPC (69%), CHOL (20%), DOPS (10%), PIP$_2$ (1%) (Supplementary Table 4). The location of the N-terminal patch and the PIP$_2$-binding site (PBS) are highlighted by gray boxes. In the upper panel, contacts for all lipids, in lower panel, only contacts with PIP$_2$ are shown. Four replicate simulations per IDR sequence were carried out for 38 μs and contact averages were calculated from the last ~28 μs of each simulation. **e** $^{31}$P NMR spectra of diC$_8$-PIP$_2$ (light blue) with increasing amounts of IDR, IDR$^{ΔN97}$ or IDR$^{Patch}$. Chemical shift changes are indicated by arrows. **f** Chemical shift perturbations of P4 and P5 lipid headgroup resonances upon addition of IDR (gray), IDR$^{ΔN97}$ (blue), or IDR$^{Patch}$ (mauve). Source data are provided as a Source Data file.

In summary, to understand TRP channel function, their often extensive IDRs cannot be ignored. Our work shows that "IDR cartography", i.e., mapping structural and functional properties onto distinct IDR regions through an integrated structural biology approach, can shed light on the complex regulation of a membrane receptor through its hitherto mostly neglected regions.

## Methods

### Antibodies and reagents
All chemicals were purchased from Sigma-Aldrich, Roth and VWR unless otherwise stated. Reagents used include HC067047 (Sigma-Aldrich, SML0143), GSK1016790A (Sigma-Aldrich, G0798), Alexa-Fluor 555 Phalloidin (ThermoFisher Scientific), $^{15}$N NH$_4$Cl and $^{13}$C$_6$-glucose (Eurisotop). DSS-H12/D12 for crosslinking was obtained from Creative Molecules Inc. Lipids were purchased from Avanti Polar Lipids and Cayman Chemicals. Antibodies used were rabbit anti-GFP (Thermo Fisher Scientific, A-11122), rabbit anti-β-actin (Cell Signaling Technology, 4967) and HRP-conjugated monoclonal mouse anti-rabbit IgG, light chain specific (Jackson ImmunoResearch, 211-032-171). Both antibodies were used at a dilution of 1:1000.

### Computational tools
Freely available computational tools were used to investigate the properties of N-terminal TRPV4 constructs. Sequence conservation was determined with ConSurf[50] (Figs. 5, 6 and S11). Overall charge ($z$) of IDR deletion constructs was determined with ProtPi (www.protpi.ch) (Fig. 5). Charge distribution $z$ for N-terminal deletion constructs was determined with www.bioinformatics.nl/cgi-bin/emboss/charge (Fig. 5). Gel densitometry analysis was carried out with ImageJ[51] (Fig. 7b). The IDR charge gradient in Fig. 7c was plotted with PepCalc tool (https://pepcalc.com/).

### Cloning, expression and purification of recombinant proteins
The DNA sequences encoding for the *G. gallus* TRPV4 N-terminal domain was cloned into a pET11a vector with an N-terminal His$_6$SUMO-tag, as described previously[29]. Human TRPV4 constructs in a pcDNA3.1 vector were commercially obtained from GenScript. Expression plasmids encoding for the isolated intrinsically disordered region (IDR), the isolated ankyrin repeat domain (ARD), N-terminal truncations (NTD$^{ΔN54}$, NTD$^{ΔN97}$, NTD$^{ΔN104}$, and NTD$^{ΔN120}$) and a peptide comprising residues 97–134 of the IDR (IDR$^{D97}$) were obtained from the NTD encoding vectors using a Gibson Deletion protocol[52]. Site-directed

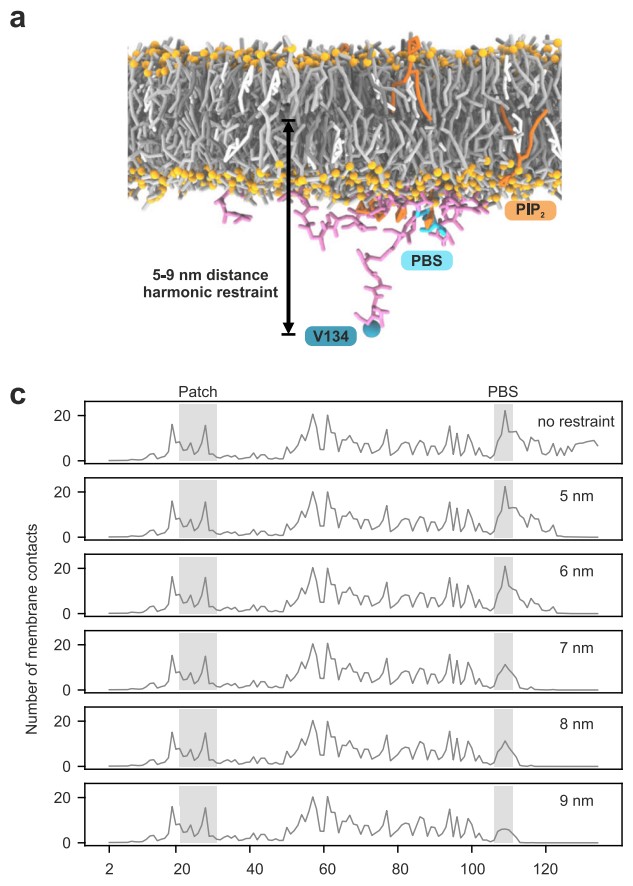

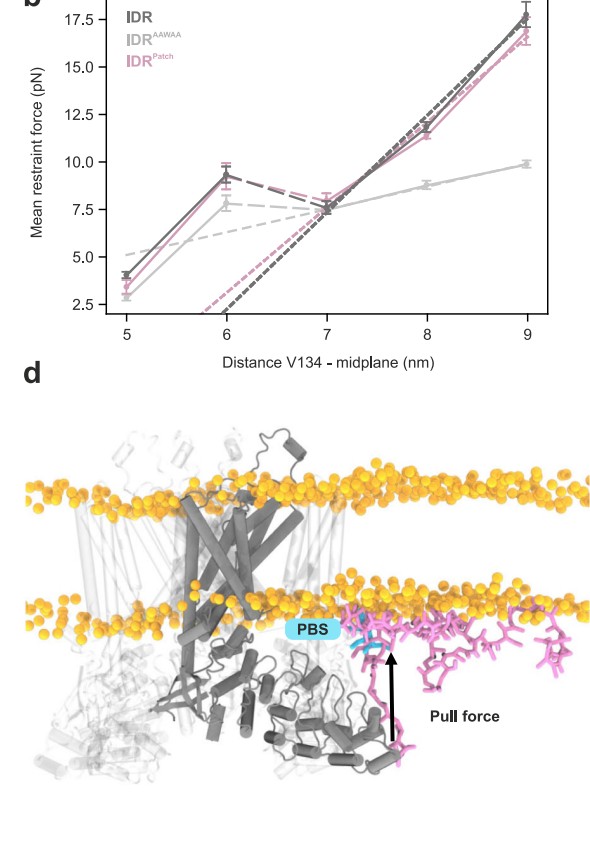

**Fig. 9 | PIP₂ binding to the TRPV4 IDR's PIP₂-binding site exerts a pulling force on the ARD. a** Coarse-grained MD simulation system setup using a lipid bilayer membrane consisting of PIP₂ (1%, dark orange) as well as POPC (69%, dark gray), DOPS (10%, light gray) and cholesterol (20%, white). Headgroup phosphates are shown as orange spheres), the IDR (pink liquorice) was kept at defined distances from the membrane midplane by its most C-terminal residue V134 (blue sphere) to emulate anchoring by the ARD. The PIP₂-binding site (PBS) is highlighted in cyan. **b** Force displacement curves from restrained simulations of TRPV4 IDR, IDR^AAWAA and IDR^Patch. Four 38 μs replicate simulations were carried out for each condition. The mean restraint force is plotted against the mean distance between residue V134 and the membrane midplane. Dotted lines show linear fits of the force contribution of the PIP₂-binding site. Averages were calculated from the last ~28 μs of each of the 4 replicate simulations per IDR genotype and per height restraint. Error bars show the standard errors of the mean (SEM) of the replicate simulations. **c** Number of membrane lipid contacts for each residue of the native IDR at a given height restraint (for results with IDR^AAWAA and IDR^Patch, see Supplementary Fig. 10c). Averages are calculated from the last 28 μs of each of the four replicate simulations. **d** Composite figure of a structure of the native IDR (from an MD simulation at a restraint distance of 7 nm) and an AlphaFold[94] model of the transmembrane core of the *G. gallus* TRPV4 tetramer. The force displacement curves in **b** indicate that the interaction of the PIP₂-binding site with the membrane exerts a pull force on the ARD N-terminus (solid arrow). Source data are provided as a Source Data file.

mutagenesis of the PIP₂-binding site (¹⁰⁷KRWRR¹¹¹ to ¹⁰⁷AAWAA¹¹¹, forward primer GTGAAAACGCAGCCTGGGCCGCGCGTGTGGTTGAAAAA CCAGTGG; reverse primer CACACGCGCGGCCCAGGCTGCGTTTTCA CCACCAATCTGT) and regulatory patch (¹⁹FPLSSLANLFE²⁹ to ¹⁹FP(AG)₅E²⁹, forward primer GATGACTCCTTCCCGGCCGGCGCGGG CGCCGGCGCGGGTGCGGGTGAGGACACCCGTCT; reverse primer CGGGGAAGGAGTCATCCCCCAGCACGTCCCC) were introduced in the abovementioned constructs by site-directed mutagenesis using polymerase chain reaction (PCR).

TRPV4 N-terminal constructs were expressed in *Escherichia coli* BL21-Gold(DE3) (Agilent Technologies) grown in terrific broth (TB) medium (or LB medium for IDR^ΔN97) supplemented with 0.04% (w/v) glucose and 0.1 mg/mL ampicillin. Cells were grown to an OD₆₀₀ of 0.8 for induction with 0.5 mM IPTG (final concentration) and then further grown at 37 °C for 3 h. ¹⁵N, ¹³C-labeled proteins were prepared by growing cells in M9 minimal medium[53] with ¹⁵N-HN₄Cl and ¹³C-glucose as the sole nitrogen and carbon sources. Cells were grown at 37 °C under vigorous shaking to an OD₆₀₀ of 0.4, moved to RT, grown to OD₆₀₀ of 0.8 for induction of protein expression with 0.15 mM IPTG

(final concentration) and then grown overnight at 20 °C. After harvest by centrifugation, cells were stored at −80 °C until further use.

All purification steps were carried out at 4 °C. Cell pellets were dissolved in lysis buffer (20 mM Tris pH 8, 20 mM imidazole, 300 mM NaCl, 0.1% (v/v) Triton X-100, 1 mM DTT, 1 mM benzamidine, 1 mM PMSF, lysozyme, DNAse, RNAse and protease inhibitor (Sigmafast)) and lysed (Branson Sonifier 250). Debris was removed by centrifugation and the supernatant applied to a Ni-NTA gravity flow column (Qiagen). After washing (20 mM Tris pH 8, 20 mM imidazole, 300 mM NaCl), proteins were eluted with 500 mM imidazole. Protein containing fractions were dialyzed overnight (20 mM Tris pH 7 (pH 8 for IDR^ΔN97), 300 mM NaCl, 10% v/v glycerol, 1 mM DTT, 0.5 mM PMSF) in the presence of Ulp-1 protease in a molar ratio of 20:1 to yield the native TRPV4 N-terminal constructs. After dialysis, cleaved proteins were separated by a reverse Ni-NTA affinity chromatography step and subsequently purified via a HiLoad prep grade 16/60 Superdex200 or 16/60 Superdex75 column (GE Healthcare) equilibrated with 20 mM Tris pH 7, 300 mM NaCl, 1 mM DTT. Pure sample fractions were flash-frozen in liquid nitrogen and stored at −20 °C until further use.

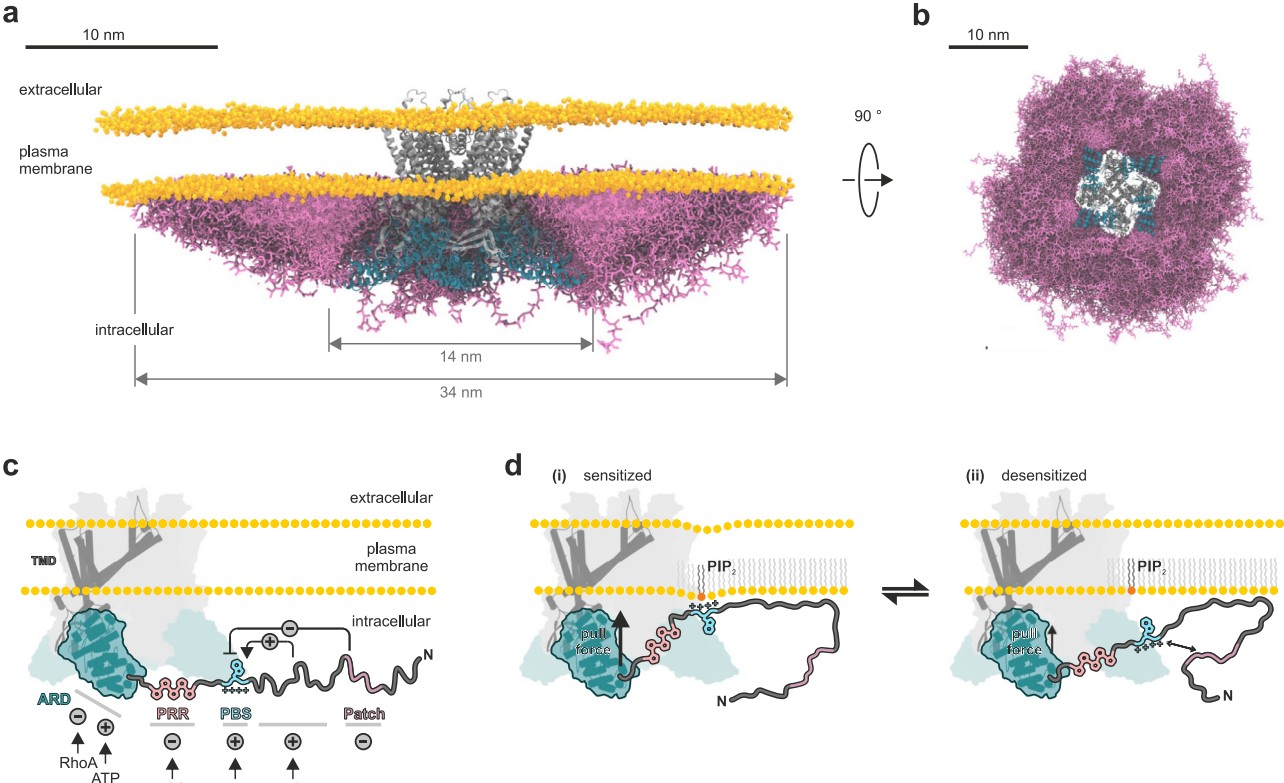

**Fig. 10 | The TRPV4 IDR forms an extensive 'belt' along the membrane plane and encodes a hierarchy of antagonistic regulatory modules. a, b** Superimposed IDR conformations from coarse-grained MD simulations (pink licorice) integrated into a full-length TRPV4 tetramer (AlphaFold[94] prediction of *G. gallus* TRPV4 transmembrane core (gray) and ARDs (cyan)) viewed from the side **a** and from the intracellular side **b**. For better visualization of the extent of the IDR 'belt', the IDR conformations of the front facing TRPV4 monomer have been deleted. For a view of the IDR conformations on a single TRPV4 subunit, see Supplementary Fig. 13 as well as Supplementary Movies 2 and 3. **c** The TRPV4 N-terminus encodes multiple antagonistic regulatory elements that regulate TRPV4 function through ligand, protein, lipid or intra-domain contacts. **d** PIP$_2$-binding site interactions with the membrane exert a pull force on the IDR C-terminus. Likewise, pulling on the IDR can lead to membrane deformation. Crosstalk between the PIP$_2$-binding site and the autoinhibitory patch modulates PIP$_2$ binding and thus IDR membrane interactions, thereby influencing channel activity.

Purified IDR$^{\Delta N97}$ was extensively dialyzed against double distilled water, lyophilized, and stored in solid form at −20 °C. Peptides could be dissolved in desired amounts of buffer to concentrations up to 10 mM.

## Western blot

HEK293T cells were cultured in Dulbecco's Modified Eagle's Medium (DMEM) supplemented with 10% (v/v) fetal calf serum (FCS) and penicillin/streptomycin at 37 °C with 6% CO$_2$. For Western blots, HEK293T cells were transfected with Lipofectamine LTX with Plus Reagent (Thermo Fisher Scientific), lysed 24 h after transfection in RIPA Buffer (150 mM NaCl, 1.0% IGEPAL CA-630, 0.5% sodium deoxycholate, 0.1% SDS, 50 mM Tris, pH 8.0, Millipore Sigma) supplemented with Halt protease inhibitor cocktail (Thermo Fisher Scientific) followed by centrifugation at 17,000 × g for 10 min. Supernatants were then removed and Laemmli sample buffer with β-mercaptoethanol was added. Protein lysates were heated for 10 min at 70 °C and resolved on 4−15% TGX gels (Bio-Rad Laboratories) and then transferred to PVDF membranes. Membranes were developed using SuperSignal West Femto Maximum Sensitivity Substrate (Thermo Fisher Scientific) and imaged using an ImageQuant LAS 4000 system (GE Healthcare).

## Analytical size-exclusion chromatography

Analytical SEC experiments were carried out at 4 °C using an NGC Quest (BioRad) chromatography system. A total of 250 µL protein at a concentration of 2−3 mg/mL was injected on a Superdex200 10/300 increase column (GE Healthcare) equilibrated with 20 mM Tris pH 7,

300 mM NaCl, 1 mM DTT via a 1 mL loop. Protein was detected by absorbance measurement at wavelengths of 230 and 280 nm.

For Stokes radius ($R_S$) determination, SEC columns were calibrated with a protein standard kit (GE Healthcare) containing ferritin ($M_W$ = 440 kDa, $R_S$ = 61.0 nm), alcohol dehydrogenase (150 kDa, 45.0 nm), conalbumin (75 kDa, 36.4 nm), ovalbumin (43 kDa, 30.5 nm), carbonic anhydrase (29 kDa, 23.0 nm), ribonuclease A (13.7 kDa, 16.4 nm), and aprotinin (6.5 kDa, 13.5 nm) whose Stokes radii were obtained from La Verde et al.[54] The SEC elution volume, $V_e$ (in mL), of the protein standards was plotted versus the log$R_S$, with $R_S$ in nm, and fitted with a linear regression (Eq. 1):

$$V_e = m \cdot \log R_S + b \tag{1}$$

where $m$ is the slope of the linear regression and $b$ the y-axis section. Equation 1 was then used to calculate the Stokes radii of the TRPV4 constructs from their respective SEC elution volumes.

## Size-exclusion chromatography multi-angle light scattering (SEC-MALS)

Multi-angle light scattering coupled with size-exclusion chromatography (SEC-MALS) of the *G. gallus* TRPV4 NTD, ARD, and IDR was performed with a GE Superdex200 Increase 10/300 column run at 0.5 mL/min on a Jasco HPLC unit (Jasco Labor und Datentechnik) connected to a light scattering detector measuring at three angles (miniDAWN TREOS, Wyatt Technology). The column was equilibrated for at least 16 hrs with 20 mM Tris pH 7, 300 mM NaCl, 1 mM DTT

(filtered through 0.1 µm pore size VVLP filters (Millipore)) before 200 µL of protein samples at a concentration of 2 mg/mL were loaded. The ASTRA 7 software package (Wyatt Technology) was used for data analysis, assuming a Zimm model[55]. The molecular weight, $M_W$, can be determined from the reduced Rayleigh ratio extrapolated to zero, $R(0)$, which is the light intensity scattered from the analyte relative to the intensity of the incident beam (Eq. 2):

$$M_W = \frac{R(0)}{K \cdot C \cdot \left(\frac{dn}{dc}\right)^2} \qquad (2)$$

Here, $c$ is the concentration of the analyte and $(dn/dc)$ is the refractive index increment, which was set to 0.185 mL/g, a standard value for proteins[56]. $K$ is an optical constant depending on wavelength and the solvent refractive index. The protein extinction coefficients at 280 nm were calculated from the respective amino acid sequences using the ProtParam tool[57].

## Circular dichroism (CD) spectroscopy

CD measurements were carried out on a Jasco-815 CD spectrometer (JascoTM) with 1 mm quartz cuvettes (Hellma Macro Cell). Proteins were used at concentrations in the range of 0.03–0.05 mg/mL in 5 mM Tris pH7, 10 mM NaCl. Spectra were recorded at 20 °C between 190 and 260 nm with 1 nm scanning intervals, 5 nm bandwidth and 50 nm/min scanning speed. All spectra were obtained from the automatic averaging of three measurements with automatic baseline correction. The measured ellipticity $\theta$ in degrees (deg) was converted to the mean residue ellipticity (MRE) via Eq. 3[58].

$$\mathrm{MRE}_\lambda = \frac{\mathrm{MRW} \cdot \theta \lambda}{10 \cdot d \cdot c} \qquad (3)$$

Here, $\mathrm{MRE}_\lambda$ is the mean residue ellipticity, and $\theta_\lambda$ is the measured ellipticity at wavelength $\lambda$, $d$ is the pathlength (in cm), and $c$ is the protein concentration (g/mL). MRW is the mean residue weight, $\mathrm{MRW} = \frac{M_W}{N-1}$, where $M_W$ is the molecular weight of the protein (in Da), and $N$ is the number of residues. For titration experiments, TRPV4 N-terminal peptides were used in a concentration of 30 µM in double distilled water in the presence of TFE (2,2,2-trifluoroethanol, 0–90% (v/v)), SDS (0.5, 1.0, 2.5, 5.0 and 8.0 mM) and liposomes (0.5 and 1.0 mM). Liposomes were prepared from POPG and POPC at a molar ratio of 1:1 as described below.

## Small angle X-ray scattering (SAXS)

SAXS experiments were carried out at the EMBL-P12 bioSAXS beam line, DESY[59]. SEC-SAXS data collection[60], $I(q)$ vs $q$, where $q = 4\pi \sin q/\lambda$; $2q$ is the scattering angle and $\lambda$ the X-ray wavelength (0.124 nm; 10 keV) was performed at 20 °C using S75 (IDR constructs) and S200 Increase 5/150 (NTD and ARD constructs) analytical SEC columns (GE Healthcare) equilibrated in the appropriate buffers (see Supplementary Tables 1 and 2) at flow rates of 0.3 mL/min. Automated sample injection and data collection were controlled using the *BECQUEREL* beam line control software[61]. The SAXS intensities were measured as a continuous series of 0.25 s individual X-ray exposures, from the continuously-flowing column eluent, using a Pilatus 6 M 2D-area detector for a total of one column volume (ca. 600–3000 frames in total). The 2D-to-1D data reduction, i.e., radial averaging of the data to produce 1D $I(q)$ vs $q$ profiles, were performed using the SASFLOW pipeline incorporating RADAVER from the ATSAS 2.8 suite of software tools[62]. The individual frames obtained for each SEC-SAXS run were processed using CHROMIXS[63]. Briefly, individual SAXS data frames were selected across the respective sample SEC-elution peaks and an appropriate region of the elution profile, corresponding to SAXS data measured from the solute-free buffer, were identified, averaged, and then subtracted to generate individual background-subtracted sample

data frames. These data frames underwent further CHROMIXS analysis, including the assessment of the radius of gyration ($R_g$) of each individual sample frame, scaling of frames with equivalent $R_g$, and subsequent averaging to produce the final 1D-reduced and background-corrected scattering profiles. Only those scaled individual SAXS data frames with a consistent $R_g$ through the SEC-elution peak that were also evaluated as statistically similar through the measured $q$-range were used to generate the final SAXS profiles. Corresponding UV traces were not measured; the column eluate was flowed directly to the P12 sample exposure unit after the small column, forging UV absorption measurements, to minimize unwanted band-broadening of the sample. All SAXS data-data comparisons and data-model fits were assessed using the reduced $\chi^2$ test and the Correlation Map, or CorMap, $p$ value[64]. Fits within the $\chi^2$ range of 0.9–1.1 or having a CORMAP $p$ values higher than the significance threshold cutoff of a = 0.01 are considered excellent, i.e., no systematic differences are present between the data-data or data-model fits at the significance threshold.

Primary SAXS data analysis was performed using PRIMUS within ATSAS 3.0.1[65]. The Guinier approximation[66] (ln$I(q)$ vs. $q^2$ for $qR_g < 1.3$) and the real-space pair distance distribution function, or $p(r)$ profile (calculated from the indirect inverse Fourier transformation of the data, thus also yielding estimates of the maximum particle dimension, $D_{max}$, Porod volume, $V_p$, shape classification, and concentration-independent molecular weight[67–69] were used to estimate the $R_g$ and the forward scattering at zero angle, $I(0)$. Dimensionless Kratky plot representations of the SAXS data ($qR_g^2(I(q)/I(0))$ vs. $qR_g$) followed an approach previously described[70]. All collected SAXS data are reported in Supplementary Tables 1 and 2.

## Ensemble modeling

The ensemble analysis of IDR, NTD, the systematic NTD IDR-deletions and/or respective IDR/NTD-PIP$_2$-binding site mutants was performed using Ensemble Optimization Method, EOM[37,38]. Briefly, 10,000 protein structures were generated for each of the respective protein constructs, where the IDR section(s) were modeled as random chains (self-avoiding walks with the confines of Ramachandran-constraints). The scattering profiles were calculated for each model within the initially generated 10,000 member ensembles. The selection of sub-ensembles describing the SAXS data, and the assessment of the $R_g$ distribution of the refined ensemble pools, was performed using a genetic algorithm based on fitting the SAXS data with a combinatorial volume-fraction weighted sum contribution of individual model scattering profiles drawn from the initial pool of structures.

## Hydrogen/deuterium exchange mass spectrometry (HDX-MS)

HDX-MS was conducted on three independent preparations of *G. gallus* TRPV4 IDR, ARD or NTD protein each, and for each of those three technical replicates (individual HDX reactions) per deuteration timepoint were measured. Preparation of samples for HDX-MS was aided by a two-arm robotic autosampler (LEAP Technologies). HDX reactions were initiated by 10-fold dilution of pre-dispensed protein sample (25 µM) with buffer (20 mM Tris pH 7, 300 mM NaCl) prepared in D$_2$O and incubated for 10, 30, 100, 1000 or 10,000 s at 25 °C. The exchange was stopped by mixing with an equal volume of pre-dispensed quench buffer (400 mM KH$_2$PO$_4$/H$_3$PO$_4$, 2 M guanidine-HCl; pH 2.2) kept at 1 °C, and 100 µL of the resulting mixture injected into an ACQUITY UPLC M-Class System with HDX Technology[71]. Non-deuterated samples were generated by a similar procedure through 10-fold dilution in buffer prepared with H$_2$O. The injected HDX samples were washed out of the injection loop (50 µL) with water + 0.1% (v/v) formic acid at a flow rate of 100 µL/min and guided over a column containing immobilized porcine pepsin kept at 12 °C. The resulting peptic peptides were collected on a trap column (2 mm × 2 cm), that was filled with POROS 20 R2 material (Thermo Scientific) and kept at 0.5 °C. After three minutes,

the trap column was placed in line with an ACQUITY UPLC BEH C18 1.7 µm 1.0 × 100 mm column (Waters) and the peptides eluted with a gradient of water + 0.1% (v/v) formic acid (eluent A) and acetonitrile + 0.1% (v/v) formic acid (eluent B) at 60 µL/min flow rate as follows: 0−7 min/95−65% A, 7-8 min/65-15% A, 8-10 min/15% A. Eluting peptides were guided to a Synapt G2-Si mass spectrometer (Waters) and ionized by electrospray ionization (capillary temperature and spray voltage of 250 °C and 3.0 kV, respectively). Mass spectra were acquired with the software MassLynx MS version 4.1 (Waters) over a range of 50 to 2000 $m/z$ in enhanced high definition MS (HDMS$^E$)[72,73] or high definition MS (HDMS) mode for non-deuterated and deuterated samples, respectively. Lock mass correction was conducted with [Glu1]-Fibrinopeptide B standard (Waters). During separation of the peptides on the ACQUITY UPLC BEH C18 column, the pepsin column was washed three times by injecting 80 µL of 0.5 M guanidine hydrochloride in 4% (v/v) acetonitrile. Blank runs (injection of double-distilled water instead of the sample) were performed between each sample. All measurements were carried out in triplicates. Peptides were identified and evaluated for their deuterium incorporation with the software ProteinLynx Global SERVER 3.0.1 (PLGS) and DynamX 3.0 (both Waters). Peptides were identified with PLGS from the non-deuterated samples acquired with HDMS$^E$ employing low energy, elevated energy and intensity thresholds of 300, 100, and 1000 counts, respectively and matched using a database containing the amino acid sequences of IDR, ARD, NTD, porcine pepsin and their reversed sequences with search parameters as follows: Peptide tolerance = automatic; fragment tolerance = automatic; min fragment ion matches per peptide = 1; min fragment ion matches per protein = 7; min peptide matches per protein = 3; maximum hits to return = 20; maximum protein mass = 250,000; primary digest reagent = non-specific; missed cleavages = 0; false discovery rate = 100. For quantification of deuterium incorporation with DynamX, peptides had to fulfill the following criteria: Identification in at least 2 of the 3 non-deuterated samples; the minimum intensity of 10,000 counts; maximum length of 30 amino acids; minimum number of products of two; maximum mass error of 25 ppm; retention time tolerance of 0.5 min. All spectra were manually inspected and omitted, if necessary, e.g., in case of low signal-to-noise ratio or the presence of overlapping peptides disallowing the correct assignment of the isotopic clusters.

Residue-specific deuterium uptake from peptides identified in the HDX-MS experiments was calculated with the software DynamX 3.0 (Waters). In the case that any residue is covered by a single peptide, the residue-specific deuterium uptake is equal to that of the whole peptide. In the case of overlapping peptides for any given residue, the residue-specific deuterium uptake is determined by the shortest peptide covering that residue. Where multiple peptides are of the shortest length, the peptide with the residue closest to the peptide C-terminus is utilized. Assignment of residues being intrinsically disordered was based on two criteria, i.e., a residue-specific deuterium uptake of >50% after 10 s of HDX and no further increment in HDX > 5% in between consecutive HDX times. Raw data of deuterium uptake by the identified peptides and residue-specific HDX are provided in Supplementary Data 1.

## Crosslinking mass spectrometry (XL-MS)

For structural analysis, 100 µg of purified protein were crosslinked by addition of DSS-H12/D12 (Creative Molecules) at a ratio of 1.5 nmol/1 µg protein and gentle shaking for 2 h at 4 °C. The reaction was performed at a protein concentration of 1 mg/mL in 20 mM HEPES pH 7, 300 mM NaCl. After quenching by addition of ammonium bicarbonate (AB) to a final concentration of 50 mM, samples were dried in a vacuum centrifuge. Then, proteins were denatured by resuspension in 8 M urea, reduced with 2.5 mM Tris(2-carboxyethyl)-phosphine (TCEP) at 37 °C

for 30 min and alkylated with 5 mM iodoacetamide at room temperature in the dark for 30 min. After dilution to 1 M urea using 50 mM AB, 2 µg trypsin (protein:enzyme ratio 50:1; Promega) were added and proteins were digested at 37 °C for 18 h. The resulting peptides were desalted by C18 Sep-Pak cartridges (Waters), then crosslinked peptides were enriched by size exclusion chromatography (Superdex 20 Increase 3.2/300, cytiva) prior to liquid chromatography (LC)-MS/MS analysis on an Orbitrap Fusion Tribrid mass spectrometer (Thermo Scientific)[74]. MS measurement was performed in data-dependent mode with a cycle time of 3 s. The full scan was acquired in the Orbitrap at a resolution of 120,000, a scan range of 400−1500 m/z, AGC Target 2.0e5 and an injection time of 50 ms. Monoisotopic precursor selection and dynamic exclusion for 30 s were enabled. Precursor ions with charge states of 3-8 and minimum intensity of 5e3 were selected for fragmentation by CID using 35% activation energy. MS2 was done in the Ion Trap in rapid scan range mode, AGC target 1.0e4 and a dynamic injection time. All experiments were performed in biological triplicates and samples were measured in technical duplicates. Crosslink analysis was done with the xQuest/xProphet pipeline[75] in ion-tag mode with a precursor mass tolerance of 10 ppm. For matching of fragment ions, tolerances of 0.2 Da for common ions and 0.3 Da for crosslinked ions were applied. Crosslinks were only considered for further analyses if they were identified in at least 2 of 3 biological replicates with deltaS <0.95 and at least one Id score ≥ 25.

## Lipid preparation

Liposomes were prepared from 1-palmitoyl-2-oleoyl-sn-glycero-3-phosphocholine (POPC) and 1-palmitoyl-2-oleoyl-sn-glycero-3-phosphoglycerol (POPG) mixed in a 1:1 (n/n) ratio in chloroform. The organic solvent was removed via nitrogen flux and under vacuum via desiccation overnight. The lipid cake was suspended in 1 mL buffer (10 mM Tris pH 7, 100 mM NaCl or 300 mM NaCl) and incubated for 20 min at 37 °C and briefly spun down before being subjected to five freeze and thaw cycles. The resulting large unilamellar vesicles (LUVs) were incubated for 20 min at 21 °C under mild shaking. To obtain a homogeneous solution of small unilamellar vesicles (SUVs), the mixture was extruded 15 times through a 100 nm membrane using the Mini Extruder (Avanti Polar Lipids). This yielded a liposome stock solution of 100 nm liposomes with 4.0 mM lipid in 10 mM Tris pH 7, 100 mM NaCl that was used immediately for measurements. POPC-only liposomes were prepared similarly.

Commercial stocks containing 1,2-dioctanoyl-sn-glycero-3-phosphocholine (diC8-PC) and 1,2-dioctanoyl-sn-glycero-3-phosphoglycerol (diC8-PG) were prepared by first removing the organic solvent via nitrogen flux and under vacuum via desiccation overnight. Afterwards, the lipids were dissolved in the appropriate amount of buffer to yield 10 mM lipid stocks. 1,2-dioctanoyl-sn-glycero-3-phospho-(1'-myo-inositol-4',5'-bisphosphate) (diC8-PI(4,5)P$_2$) was purchased as a powder and directly dissolved in the appropriate amount of buffer.

## Liposome sedimentation assay

Liposomes were prepared as described above. Proteins and liposomes in 20 mM Tris pH 7, 100 mM NaCl were mixed to a final concentration of 2.5 µM protein and 2 mg/mL lipid. After incubation at 4 °C for 1 h under mild shaking, an SDS-PAGE sample of the input was taken. The mixture was then centrifuged at 70,000 g for 1 h at 4 °C. SDS-PAGE samples (15 µL) were taken from both the supernatant and the pellet resuspended in assay buffer. Control samples without liposomes were run in parallel to verify the protein stability under the experimental conditions. The protein distribution between the pellet and supernatant fractions was determined by running an SDS-PAGE and densitometrically analyzing the bands using imageJ[51]. Sedimentation assays were carried out three times for each protein liposome mixture and protein only control sample. Error bars were calculated as the standard deviation from the mean value of three replicates.

## Nuclear magnetic resonance (NMR) spectroscopy

Backbone assignments of native $^{13}$C, $^{15}$N-labeled *G. gallus* IDR have been reported by us previously[5]. For complete backbone assignments of IDR$^{AAWAA}$ and IDR$^{Patch}$, $^{15}$N, $^{13}$C-labeled proteins were prepared. Backbone and side chain chemical shift resonances were assigned with a set of band-selective excitation short-transient (BEST) transverse relaxation-optimized spectroscopy (TROSY)-based assignment experiments: HNCO, HN(CA)CO, HNCA, HN(CO)CA, HNCACB. Additional side chain chemical shift information was obtained from H(CCCO)NH and (H)CC(CO)NH experiments.

For titrations with lipids, or comparison of chemical shifts between constructs, $^{15}$N-labeled IDR and NTD variants were prepared. All NMR spectra were recorded at 10 °C on 600 MHz to 950 MHz Bruker AvanceIII HD NMR spectrometer systems equipped with cryogenic triple resonance probes. For peptide and lipid titration experiments, a standard [$^1$H,$^{15}$N]-BEST-TROSY pulse sequence implemented in the Bruker Topspin pulse program library was used. Solutions with 100 μM of $^{15}$N-labeled IDR constructs in 10 mM Tris-HCl pH 7, 100 mM (or 300 mM) NaCl, 1 mM DTT, 10% (v/v) D$_2$O were titrated with lipid from a concentrated stock solution. The chemical shifts were determined using TopSpin 3.6 (Bruker) The $^1$H and $^{15}$N weighted chemical shift differences observed in [$^1$H, $^{15}$N]-HSQC spectra were calculated according to Eq. 4[76]:

$$\Delta\delta = \sqrt{\Delta\delta_H^2 + \left(\frac{\Delta\delta_N}{6.5}\right)^2} \qquad (4)$$

Here, $\Delta\delta_H$ is the $^1$H chemical shift difference, $\Delta\delta_N$ is the $^{15}$N chemical shift difference, and $\Delta\delta$ is the $^1$H and $^{15}$N weighted chemical shift difference in ppm.

For NMR titrations of $^{15}$N-labeled TRPV4 IDR with liposomes (SUVs) where line broadening instead of peak shifts was observed, the interaction of the reporter with liposomes was quantified using the peak signal loss in response to liposome titration. The signal loss at a lipid concentration $c_i$ was calculated as the relative peak signal decrease rel. $\Delta I$ according to Eq. 5.

$$\text{rel. } \Delta I = \frac{I_i - I_0}{I_0} \qquad (5)$$

Here, $I_O$ is the peak integral in the absence of SUVs, and $I_i$ is the peak integral in the presence of a lipid concentration $c_i$.

$^{31}$P{$^1$H} NMR spectra were recorded at 25 °C on a 600 MHz Bruker AvanceIII HD NMR spectrometer. DiC$_8$-PI$_{(4,5)}$P$_2$ was used at 500 μM in 10 mM Tris pH 7, 100 mM NaCl, 10% (v/v) D$_2$O and titrated with protein from a concentrated stock solution.

Standard NMR pulse sequences implemented in Bruker Topspin library were employed to obtain $R_1$, $R_2$ and $^{15}$N,{$^1$H}-*NOE* values. Longitudinal and transverse $^{15}$N relaxation rates ($R_1$ and $R_2$) of the $^{15}$N-$^1$H bond vectors of backbone amide groups were extracted from signal intensities ($I$) by a single exponential fit according to Eq. 6:

$$I = I_0 e^{-tR_{1/2}} \qquad (6)$$

In $R_1$ relaxation experiments, the variable relaxation delay $t$ was set to 1000 ms, 20 ms, 1500 ms, 60 ms, 3000 ms, 100 ms, 800 ms, 200 ms, 40 ms, 400 ms, 80 ms, and 600 ms. In all $R_2$ relaxation experiments, the variable loop count was set to 36, 15, 2, 12, 4, 22, 8, 28, 6, 10, 1, and 18. The length of one loop count was 16.96 ms. The variable relaxation delay $t$ in $R_2$ experiments is calculated by length of one loop count times the number of loop counts. The inter-scan delay for the $R_1$ and $R_2$ experiments was set to 3 s. The {$^1$H}–$^{15}$N steady-state nuclear Overhauser effect measurements ({$^1$H},$^{15}$N-*NOE*) were obtained from separate 2D $^1$H-$^{15}$N spectra acquired with and without continuous $^1$H saturation, respectively. The {$^1$H},$^{15}$N-*NOE* values were determined by taking the ratio of peak volumes from the two spectra, {$^1$H},$^{15}$N-*NOE* = $I_{sat}/I_O$, where $I_{sat}$ and $I_O$ are the peak intensities with and without $^1$H saturation. The saturation period was approximately $5/R_1$ of the amide protons.

## Tryptophan fluorescence spectroscopy

All tryptophan fluorescence measurements were carried out in 10 mM Tris pH 7, 100 mM NaCl buffer on a Fluro Max-4 fluorimeter with an excitation wavelength of 280 nm and a detection range between 300 nm and 550 nm. The fluorescence wavelength was determined as the intensity-weighted fluorescence wavelength between 320 and 380 nm (hereafter referred to as the average fluorescence wavelength) according to Eq. 6:

$$\langle\lambda\rangle = \frac{\sum_{i=320\,nm}^{n=380\,nm} I_i \cdot \lambda_i}{\sum_{i=320\,nm}^{n=380\,nm} I_i} \qquad (6)$$

Here, $\langle\lambda\rangle$ is the average fluorescence wavelength and $I_i$ is the fluorescence intensity at wavelength $\lambda_i$.

To monitor liposome binding by tryptophan fluorescence, fluorescence emission spectra were recorded in the presence of increasing lipid concentrations. Lipids were prepared as SUVs with 100 nm diameters as described above. The protein concentrations were kept constant at 5 μM. The protein-liposome mixtures were incubated for 10 min prior to recording the emission spectra. For each sample, at least three technical replicates were measured. The tryptophan fluorescence wavelength at each lipid concentration was quantified by determining the average fluorescence wavelength using Eq. (6). The dissociation constants, $K_d$, of the protein liposome complexes were determined by plotting the changes in $\langle\lambda\rangle$ against the lipid concentration $c$ and fitting the data with a Langmuir binding isotherm (Eq. (7)):

$$\Delta\langle\lambda\rangle = \frac{\Delta\langle\lambda\rangle_{max} \cdot c}{K_d + c} \qquad (7)$$

Here, $\Delta\langle\lambda\rangle$ describes the wavelength shift between the spectrum in the absence of lipids and a given titration step at a lipid concentration $c$. $\Delta\langle\lambda\rangle_{max}$ indicates the maximum wavelength shift in the saturation regime of the binding curve. Under the assumption that proteins cannot diffuse through liposome membranes and therefore can only bind to the outer leaflet, the lipid concentration $c$ was set to half of the titrated lipid concentration.

## Calcium imaging

MN-1 cells were transfected with GFP-tagged TRPV4 plasmids using Lipofectamine LTX with Plus Reagent. Calcium imaging was performed 24 h after transfection on a Zeiss Axio Observer.Z1 inverted microscope equipped with a Lambda DG-4 (Sutter Instrument Company, Novato, CA) wavelength switcher. Cells were bath-loaded with Fura-2 AM (8 μM, Life Technologies) for 45–60 min at 37 °C in calcium-imaging buffer (150 mM NaCl, 5 mM KCl, 1 mM MgCl$_2$, 2 mM CaCl$_2$, 10 mM glucose, 10 mM HEPES, pH 7.4). For hypotonic saline treatment, one volume of NaCl-free calcium-imaging buffer was added to one volume of standard calcium-imaging buffer for a final NaCl concentration of 70 mM. For GSK101 treatment, GSK101 was added directly to the calcium imaging buffer to achieve 50 nM final concentration. Cells were imaged every 10 s for 20 s prior to stimulation with hypotonic saline or GSK101, and then imaged every 10 s for an additional 2 min. Calcium levels at each time point were computed by determining the ratio of Fura-2 AM emission at 340 nm divided by the emission at 380 nm. Data were expressed as raw Fura ratio minus background Fura ratio.

## Atomistic molecular dynamics (MD) simulations

We performed MD simulations both of the isolated ARD in solution and of the complete TRPV4 channel core domain in a lipid bilayer using Gromacs versions 2020.3[77] (see also Supplementary Fig. 15, Supplementary Table 5). All production simulations used an integration timestep of 2 fs and were performed in the NPT ensemble. Constant temperature and pressure were maintained with the velocity-rescale[78] thermostat with a coupling time of 1 ps and with the Parrinello-Rahman barostat[79] with a reference pressure of 1 bar, a coupling time of 5 ps, and a compressibility factor of $4.5 \times 10^{-5}$ bar$^{-1}$. Pressure coupling was handled isotropically for the ARD in solution and semi-isotropically (with coupled x and y dimensions) for the membrane-embedded core. In all simulations, we used the particle-mesh Ewald (PME) algorithm[80]. We applied a pair-interaction cut-off distance of 1.2 nm for real-space electrostatic interactions and van-der-Waals interactions. We used the LINCS algorithm[81] to constrain the length of bonds involving hydrogen atoms.

**ARD in solution.** MD simulations of the ARD in solution were initiated from the *G. gallus* ARD crystal structure (PDB: 3W9G; residues 135-382). We modified both termini with capping groups (acetylated N-terminus, methylamidated C-terminus), placed the ARD in a rhombic dodecahedral box, and solvated the box with amber99SB-disp water and 150 mM NaCl. To amplify possible dynamic motions, we used the Amber99sb-disp[82] force field. The potential energy of the system was minimized using a steepest descent algorithm for 5000 steps with enabled positional restraints that maintain the initial protein structure. We then performed individual 125 ps long equilibration simulations for each of the target temperatures (38, 42, 52, and 62 °C) in the canonical (NVT) ensemble, each starting with individual Maxwell-Boltzmann velocities. During these initial equilibrations, we used a time step of 2 fs and separate velocity-rescale thermostats[78] for the protein and the solvent, each with a time constant of 1 ps. For each of the different temperatures we then performed at least 2.7 µs of simulations, of which we treated the first 200 ns as equilibration phase. Only the latter at least 2.5 µs were used for analysis. To calculate the root-mean-squared fluctuation (RMSF), we aligned each frame with the alpha carbon atoms of the helical elements of the ARD.

**TRPV4 core in the membrane.** For the MD simulations of the full TRPV4 core, we used AlphaFold2 multimer[83] to predict the core region (residues 134 to 771) of *G. gallus* TRPV4. Using CHARMM-GUI[84], we then placed the resulting homo-tetramer in a 20 × 20 nm POPC bilayer and solvated the system with TIP3-P water and 150 mM NaCl. We used the Charmm36m[85] force field. The core system was steepest-descent energy minimized for 5000 steps. Subsequently, we equilibrated the system first in the canonical (NVT) ensemble and with an integration time step of 1 fs for 250 ps and then in the constant-pressure (NPT) ensemble with a time step of 2 fs for 1.625 ns. For these equilibrations, the protein, the membrane and the solvent were individually weakly coupled every picosecond to a heat bath with a constant temperature of 37 °C using the Berendsen algorithm[86]. For the NPT equilibrations, we additionally used the Berendsen algorithm to maintain a constant pressure of 1 bar in our simulation system. During equilibration, initial position restraints of non-hydrogen protein atoms (4000 kJ mol$^{-1}$ nm$^{-2}$ and 2000 kJ mol$^{-1}$ nm$^{-2}$ for backbone and side chains, respectively), membrane phosphate z-positions (1000 kJ mol$^{-1}$ nm$^{-2}$) and membrane dihedral angles (1000 kJ mol$^{-1}$ nm$^{-2}$) were gradually released. After the equilibration simulations, we performed three 1-µs long replicate production simulations that were started with independent initial velocities according to the Maxwell-Boltzmann distribution. The first 100 ns of each replica were treated as equilibration and hence not used for analysis. To calculate the RMSF of the ARD domain along the vertical axis, we aligned the alpha carbon atoms of the transmembrane domains of TRPV4 for each frame. We then calculated the normalized

principal axis of the channel (cross product of two vectors connecting F457 of opposite subunits), projected the positions of all atoms in the ARD domain onto it, and calculated the RMSFs as the square root of the variance of the motion along the axis. Lastly, we averaged over the four subunits and calculated the standard error of the mean (SEM). Assuming that the four ARDs fluctuate independently, the SEM was calculated as the Bessel-corrected standard deviation of the mean divided by the square root of the number of subunits.

## Coarse-grained molecular dynamics (MD) simulations

All simulations were performed using Gromacs 2020.3[77] and the MARTINI2.2 forcefield[87,88] with rescaled protein-protein interactions to better represent the disordered nature of the IDR[89]. The scaling factor was set to $\alpha = 0.87$, which best describes the measured $R_g$ distribution of the native IDR (Supplementary Fig. 14). Protein-membrane interactions were not rescaled. All production simulations were performed with a 20 fs integration timestep. For equilibration simulations of systems with only protein, water, and NaCl ions present, a 40 fs timestep was used. A temperature of 37 °C was maintained with thermostats acting on protein, membrane, and solvent (water and ions) individually. The Berendsen thermostat[86] was used for equilibration simulations and the v-rescale thermostat[78] for production simulations, in both cases with characteristic times of 1 ps. A pressure of 1 bar was established with a semi-isotropic barostat (with coupled x and y dimensions) in simulations with a membrane present and with an isotropic barostat otherwise. We employed the Berendsen barostat[86] for equilibration simulations and switched to the Parrinello-Rahman barostat[79] for production simulations, always using a 20 ps time constant and a compressibility factor of $3 \times 10^{-4}$ bar$^{-1}$. Bond constraints were maintained using the LINCS algorithm[81]. To alleviate unequal heating of different lipid types, we increased the default LINCS order to 8[90]. Electrostatic interactions and van-der-Waals interactions were cut-off at 1.1 nm. All simulations were performed with an increased cut-off distance of 1.418 nm for the short-range neighbor list. Replicates were started from the same equilibrated structures, but their initial velocities were independently drawn from the Maxwell-Boltzmann distribution for each replicate simulation.

To set up the simulation systems, native IDR, IDR$^{AAWAA}$ and IDR$^{Patch}$ were modeled as disordered coils with atomistic resolution using the VMD molefacture plugin[91], converted to coarse-grained topologies using the martinize.py script (version 2.6), and placed in a 30 × 30 × 30 nm$^3$ box with solvent and 150 mM NaCl (see also Supplementary Fig. 15). The systems were then energy minimized using a steepest descent algorithm for 3000 steps. Subsequently, the systems were equilibrated for 20 ns with downscaled protein-protein interactions ($\alpha = 0.3$; only used during this step) to generate a relatively open initial IDR structure. The protein structure was then extracted and placed in a random position in the water phase of a 20 × 20 × 20 nm$^3$ box that contained a preequilibrated patch of a membrane modeled after the inner leaflet of the plasma membrane (see Supplementary Table 4 for membrane composition). Steepest descent energy minimization for 1000 steps was followed by MD equilibration for 100 ns. To probe the effect of IDR positioning in the full TRPV4 assembly, we set up simulations in which the backbone bead of V134 (C-terminus of the IDR) was harmonically restrained to a height of |z(V134)-z(membrane)| = 5, 6, 7, 8, or 9 nm over the midplane of the membrane with a force constant of 1000 kJ mol$^{-1}$ nm$^{-2}$. Here, z(membrane) is the center of mass of the membrane. Five sets of simulations (comprising four replicates each) were carried out for each of the three constructs using the Gromacs pull code[77] (Supplementary Table 5). To emulate the NMR experiments, we also performed four replicate MD simulations for each IDR construct without distance restraint. Each replicate was simulated for approximately 38 µs. After 10 µs full membrane binding was obtained in all systems and only the following approximately 28 µs of each replicate

were considered for analysis. In one simulation with IDR[Patch] restrained at 9 nm, the IDR reached over the periodic boundary to the other face of the membrane. This simulation was hence not included in any analysis. VMD[91] was used for visual analysis and rendering. All analyses were carried out with python scripts.

## Reporting summary

Further information on research design is available in the Nature Portfolio Reporting Summary linked to this article.

## Data availability

The NMR backbone assignment of the *G. gallus* TRPV4 N-terminal intrinsically disordered region has been deposited in the BioMagResBank (www.bmrb.io) under the accession number 51172. The XL-MS and HDX-MS data have been deposited to the ProteomeXchange Consortium via the PRIDE partner repository[92] with the project accession numbers PXD038153 and PXD041067, respectively. A spreadsheet with a summary of the conditions used for HDX-MS analyses and a full list of the peptides obtained for different TRPV4 protein constructs is available in Supplementary Data 1. The SAXS data have been deposited in the SASBDB under the accession numbers SASDQE8 (ARD), SASDQF8 (NTD), SASDQG8 (NTD[AAWAA]), SASDQH8 (NTD[ΔN54]), SASDQJ8 (NTD[ΔN97]), SASDQK8 (NTD[ΔN104]), SASDQL8 (NTD[ΔN120]), SASDQM8 (IDR), SASDQN8 (IDR[AAWAA]). Source data are provided with this paper. The high-resolution protein structures of the *G. gallus* TRPV4 ankyrin repeat domain used in this study is available in the protein database under the accession code 3W9G. Source data are provided with this paper.

## Code availability

Analysis scripts for molecular dynamics simulations along with the raw trajectory data are deposited in a zenodo repository (https://doi.org/10.5281/zenodo.7957940).

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

## Acknowledgements

We acknowledge the generous support by the beamline staff scientists at EMBL/P12, and the Centre for Biomolecular Magnetic Resonance (BMRZ), Goethe University Frankfurt, funded by the state of Hesse. We thank Sabine Häfner for technical support, Rupert Abele and Andreas Schlundt for help with SEC-MALS and SAXS measurements, Marta Bogacz and Lisa Pietrek for fruitful discussions. Access to beamline P12, EMBL (DESY), Hamburg was made available via iNEXT-ERIC, BAG proposal #SAXS-1106 "Conformational dynamics and equilibria in regulatory multi-domain proteins, RNAs and their complexes" (to U.A.H.). B.G. acknowledges a PhD fellowship of the Max Planck Graduate Center (MPGC). This project received funding from the NIH (NINDS K08 NS102509 to BAM, R35 NS122306 to C.J.S.), the Muscular Dystrophy Association (project 629305 to C.J.S.), the core facility for Interactions, Dynamics and Macromolecular Assembly (W.S., project 324652314 to Gert Bange, Marburg), the Max Planck Society (S.L.S., A.C.C., and G.H.), and the Deutsche Forschungsgemeinschaft (DFG, German Research Foundation) through grant STE 2517/5-1 (to F.S.), the collaborative research center 1507 "Membrane-associated Protein Assemblies, Machineries, and Supercomplexes" – Project ID 450648163 (to U.A.H. and G.H.) and the Cluster of Excellence "Balance of the Microverse" EXC 2051—Project-ID 390713860 (to U.A.H.). U.A.H. acknowledges an instrumentation grant for a high-field NMR spectrometer by the REACT-EU EFRE Thuringia (Recovery assistance for cohesion and the territories of Europe, European Fonds for Regional Development, Thuringia) initiative of the European Union.

## Author contributions

Conceptualization: B.G., U.A.H., analysis: B.G., C.W. B.A.M., S.L.S., J.J., A.C.C., C.M.J., W.S., G.H., U.A.H.; investigation: B.G., C.W. B.A.M., S.L.S., J.J., F.T., S.A.M., J.N., A.C.C., J.K.D., C.M.J., W.S.; writing—original draft: B.G., U.A.H., writing—review and editing: B.G., C.W., S.L.S., G.H., U.A.H.; visualization: B.G., C.W., S.L.S., A.C.C., U.A.H.; supervision: B.A.M., F.S., C.J.S., G.H., U.A.H.; funding acquisition: B.A.M., W.S., C.J.S., G.H., U.A.H. All authors read and approved the final version of the manuscript.

## Funding

## Competing interests

The authors declare no competing interests.
