## [Peer Review File · Nature Communications]

Crosstalk between regulatory elements in the disordered TRPV4 N-terminus modulates lipid-dependent channel activityREVIEWER COMMENTS

Reviewer #1 (Remarks to the Author):

In this work Goretzki, et al. studied thoroughly the role of the N-terminal intrinsically disordered region of a TRPV4 channel to understand its functional role in the context of the full length protein. This is important because most channels are rich in very long IDR that are generally not considered in structural and mechanical studies.

The conclusions of the work are that this region perform a regulatory function throughout a number of mechanisms. To reach this conclusion the authors designed and performed a number of experiments in vitro, in cell and in silico. The final picture they provide is generally consistent and robust and it is likely the most comprehensive mechanical study of a full length channel.

While the work is definitely interesting and provide relevant new knowledge the paper is not always equally well crafted in its different parts and would benefit from some revisions. To me for example was very difficult to follow the accumulation of data and hypothesis tested in different directions and sometimes disconnected one from the other. One possibility could be to try to summarise all/most hypothesis at the beginning (most comes from the NMR measures) and then discussing and testing them later. It would be easy to know from the beginning what is the overall hypothesis and then see the tests, possibly in this way many section of the results could be merged and reorganised moving some data in the supporting materials and focusing more only on a key subset of them. Furthermore, the authors often used the word "integrated" to underline the fact that they used a number of techniques to investigate their system. In my opinion the word is misused because the data generated by the work are never actually integrated together in the spirit of integrative structural biology technique (cf. 10.1016/j.cell.2019.05.016), but are simply interpreted altogether.

A point-by-point review follows where I avoid discussing in-cell experiments because they do not fall in my expertise.

* Structural ensemble of the TRPV4 N-terminal intrinsically disordered region: the authors combined SEC, SEC-MALS, CD, NMR, SEC-SAXS and ensemble modelling based on SAXS for the IDR region alone and for the full N-terminal cytosolic region of the gallus gallus protein. These data indicate that the IDR is disordered, that there are transient interactions between the IDR and the folded region (e.g. for residues ~20-35 and ~55 to 115) and that, interestingly, there is also some dynamic in the folded region itself. Here my concern is about the complete lack of discussion of possible secondary structure content of the IDR region, this may be relevant to understand and model better the interaction of specific IDR elements with the reminder of the protein as well as with the membrane. The authors should try to increase the resolution of their model ideally experimentally or by simulations and/or secondary structure predictions.

* Structural dynamics of the TRPV4 ARD: Combining HDX-MS, SAXS and NMR for the folded ARD domain the authors demonstrate that this undergo a significant conformational dynamics in the peripheral ankyrin repeats. My concern here is that this is a very interesting result that is not contextualised in the overall picture. A structural model for the conformational dynamics is missing but more importantly the role of this conformational dynamics in the overall context of the paper is neither further investigated nor discussed.

* Long-range TRPV4 NTD interactions center on the PIP2-binding site: using cross-linking experiments the authors find a long range interaction between the central region of the IDR domain and the PIP2-binding site at its C-terminus, furthermore they also found inter domain interactions between the IDR and ARD domains present both when the two constructs are studied as isolated domains (Fig. 3). This finding was also verified using the intrinsic fluorescence of the solely W residue of the NTD region that is the center of the PIP2 binding site. Multiple constructs are used to show that extending the PIP2 binding site towards both the N-terminus and the C-terminus result in a decreased fluorescent emission indicative of long range interactions. This result is very elegant and robust.

* The PIP2-binding site promotes compact NTD conformations: by mutating the PIP2 binding site from KRWRR → AAWAA and comparing the multiple constructs introduced previously the authors suggest that electrostatic is the driving force for the interaction between the PIP2 binding site and both the N-terminal part of the IDR and the ARD region. The data are convincing but they do not exclude transient interaction based on secondary structure complementarity. If electrostatic is the key would not be possible to study the unmutated constructs as a function of the ionic strength? The same observation holds for the following section (“Competing attractive and repulsive interactions between distinct IDR regions govern the NTD structural ensemble”)^[1]_{SEP}

* The IDR N-terminus attenuates IDR lipid binding: the authors performed NMR, sedimentation experiments and MD simulations to investigate the interaction between the IDR and the membrane. What they found is that while the PIP2 binding site is the most important interaction site with the membrane, some interaction is also due to the central part of the sequence while the N-terminus, that was previously found to interact with the PIP2 binding site, decreases the binding interaction with the membrane likely being in competition for binding the PIP2 site. Furthermore the interaction is prevalently with negatively charged components of the membrane. The last part of the section on the possible role of the central region of the IDR domain is speculative and may be moved to discussion.^[1]_{SEP}

* A conserved patch in the IDR N-terminus mediates transient long-range interactions and autoinhibits TRPV4: here is tested a putative interaction between a conserved patch at the N-terminus and the PIP2 binding site, but all observations comes only from the observations of NMR peaks broadening in different constructs. Again the possible role of local secondary structures is not considered as well as the role of salt-concentration is not tested. This section is very speculative in comparison to other part of the paper. In the following section is then tested the possible competition between membrane and N-terminus patch in binding the PIP2 binding site in the context of channel inhibition. These two sections are a typical example of a case were the story is very difficult to follow. It would have been much easier to start from the end instead than following what may have been the consecutiveness of the experiments.^[1]_{SEP}

* Membrane-bound PIP2-binding site exerts a pull force on the ARD: here an hypothesis about a mechanical role of PIP2 binding is built using MD simulations. The authors observe that the PIP2 binding site bound to the membrane may exert a mechanical strain on the ARD domain and this may be the way this information then affect the channel. This is a very interesting hypothesis that brings the paper to conclusion and opens to something new to test, I guess that the authors should try to test this experimentally for example by inserting a very flexible sequence between the C-terminus and the PIP2 binding site of the IDR or mutating the polyproline region that may act as a spring. This may also be a point to be connected with the internal conformational dynamics of the ARD region only briefly discussed at the beginning.

* An integrated structural model of the TRPV4 N-terminal ‘belt’ : this last section is a puzzle to me in the sense that there isn’t any integrated structural model/ensemble produced in the paper, there is no hypothesis an no test so it is not clear which result is presented. I would either remove this part or move it to the discussion.

Reviewer #2 (Remarks to the Author):

The manuscript by Goretzki et al. describe a multi-faceted study of the N-terminal domain of the TRPV4 channel consisting of an ankyrin repeat domain and an intrinsically disordered tail. The authors use an impressive range of experimental techniques – NMR, SAXS, hydrogen-deuterium MS, cross-linking MS as well as functional assays in a cellular system. The structural data is integrated via MD simulations. The experiments seem well-done and are analyzed appropriately. This is really as well as one can do for structural characterization of these kinds of proteins and a good example of the power of integrative structural biology. That advances significantly on the status quo and propose a plausible mechanism for how the NTD exerts its regulation of the channel via a pulling force. As such,

I have no major concerns regarding publication of this manuscript.

Reviewer #3 (Remarks to the Author):

The manuscript reports a structural investigation of TRPV4, which is a protein that contains both structured regions and intrinsically disordered regions (IDR). By engaging multiple structural biology tools, the authors were able to reveal the importance and the contribution of the largely overlooked intrinsically disordered domain to the function of TRPV4. Moreover, this is also a nice example demonstrating the importance of utilizing an integrated approach to solve critical structural biology questions. Regarding the mass spectrometry experiments and results of this manuscript, here are some comments that need authors' attention.

1. Authors utilize HDX as a tool to demonstrate the structural dynamics of the protein, highlighting the highly dynamic IDR of the protein by the high deuterium uptake at 10s. Some of the regions in the IDR are not covered. The authors should comment on the low sequence coverage of the IDR in the manuscript. The low sequence coverage might be related to the stringent criteria for the peptide identification and the intrinsic limitation of online pepsin digestion, and the audience will have a better understanding of the limitations of HDX-MS experiments if the authors can call this out.

2. In page 16, authors mentioned that the HDX experiment was by LEAP automation system. To the best of my understanding the lower limit of the H/D exchange time in the LEAP system setting is 10s, but the 10s is not accounting for the mixing time that the system is taking. Upon initiating the H/D exchange, the syringe would inject the protein into deuterated buffer as described by the authors, following by a mixing step that can take up to 60s (depending on the setting of liquid drawing speed during mixing). The H/D exchange time is counted starting from the end of the mixing step, but the actual exchange begins at the time when mixing is initiated. This delay (because of the mixing step) has a more significant impact on shorter H/D exchange time points (10s) as compared with longer points (1000s). Such a limitation needs to be verified by revisiting the experimental settings, and needs to be disclosed in the experimental section if confirmed.

3. In page 17, authors mentioned that the cross-linked peptides were enriched by SEC before submitted to LC-MS analysis. Typically, enrichment is performed before digestion, but in this case, if the primary focus is on intra-molecular cross-link, this strategy may not work. However, after digestion, it might be hard to distinguish cross-linked peptides versus non-crosslinked but longer peptides. I wonder what is the principle when using SEC to enrich cross-linked species at the peptide level, and how do authors make sure that this enrichment is non-discriminative. It will be nice if the authors can include more details on this important step.

Reviewer #4 (Remarks to the Author):

In this report by Goretzki and colleagues, the authors use an ensemble of approaches to characterize the role of the intrinsically disordered N-terminal tail of TRPV4. Previous studies identified two regulatory motifs in the N-terminal domain, a proline-rich domain and a PIP2-binding site. Here, the authors identify an autoinhibitory patch in the N-terminal domain that can compete with PIP2 for the PIP2 binding site and that disruption of the interaction between the PIP2 binding site and the membrane lessens the force exerted on the structured domains of TRPV4 by the N-terminal domain. Based on these analyses, the authors propose a hierarchical model for disparate stimuli can regulate the activity of TRPV4. While these results are interesting and can potentially improve the understanding of this channel, the model is inadequately validated, and direct functional assessments of the model should be performed prior to publication.

Major comments:

1. As part of this work, the authors use a wide array of approaches. However, a more complete description of the various approaches and especially a description of their limitations would be helpful for the broad readership of Nature Communications.
2. The Ca²⁺ imaging data is difficult to interpret due to the large differences in the expression of mutant channels as shown in Figure S7. How was the difference in expression controlled for in these assays?
3. There also appears to be several GFP-positive fragments in each lane. Do the lower bands correspond to degradation products and how would degradation influence the activity of the channel?
4. The authors propose that the NTD regulates TRPV4 through a hierarchical network of lipid-dependent interactions. However, this model is not directly tested in manuscript. Electrophysiological recordings of excised patches from cells expressing wild-type and mutant channels in the presence and absence of PIP2 and other anionic lipids is necessarily to demonstrate the validity of the proposed model.
5. Throughout the manuscript the authors employ deletion constructs to analyze the role of specific domains. However, by deleting large regions of the protein, they exclude the possibility that non-specific interactions can contribute to the regulation of the channel. The authors should repeat some of the experiments with constructs in which the sequences of the regions being tested are scrambled rather than deleted to allow discrimination between specific and non-specific effects.

Reviewer #5 (Remarks to the Author):

Goretski et al.

Crosstalk between regulatory elements in the disordered TRPV4 N-terminus modulates lipid-dependent channel activity

The manuscript submitted by Goretski et al. aims to characterize the regulatory elements within the TRVP4 IDR completely. Regulation of many classes of proteins (ion channels included) by attached or interacting IDRs is critical to many functions. Despite this, only a handful of frequently occurring IDR regulatory motifs are well understood. I am impressed by the work done in this manuscript to understand the structure/disorder function relationship in the TRVP4 ion channel. I would go as far as to say that this manuscript puts on display one of the most rigorous structural analyses of an IDR I have encountered. The authors deploy nearly all of the canonical experimental techniques used to study IDRs (NMR, SAXS, HDX, SEC-MALS, CD spec, simulation) and combine these with functional assays in a cellular background. I have no hesitation in recommending this manuscript. The only additional experiments I would consider suggesting would be far outside the scope and only occur to me due to how complete the manuscript already is (FRET experiments in the full-length constructs in a membrane environment?).

I have a few suggestions that may improve the final product.

The manuscript does display a dizzying array of techniques. I believe it would help the reader if the experimental plan and the significant conclusions were clearly on display in the introduction.

Some analyses are a bit redundant. There is little need to show the Pr distribution, and the EOM fits. They provide more or less the same information, given that the underlying conformations in the EOM ensembles are not interpretable (not that there is an attempt to interpret them here). I would forgo either example in favor of analysis with CG or atomistic simulations, but that is nitpicking.

Similarly, I think that panels b-d in figure one are a bit redundant. I believe information about the quality of the protein can be moved to the supplemental information.

The purples and greys in figure 1 are a bit hard to distinguish (particularly when printed)

The classifications of protein flexibility used in fig S1d are unnecessarily fine-grained and muddy the interpretation. The definitions of Uversky are up for debate. It would be more clear to simply label known reference states such as "folded," "theta-chain," etc. The SARW reference state is already used as a point of comparison for the SAXS data.

The fact that a D_{max} of $\sim 11.5\text{nm}$ is required to fit data from a globular protein indicates that larger structures are present in the data. I would also say that in the norm Kratky plot (S3e), the ARD might not look like a pure globule. To my eyes, the peak is located at $qR_g > \sqrt{3}$. I suspect that adding the reference point ($\sqrt{3}$, 1.1) to the Kratky plot would further support your conclusions.

Reviewer #1 (Remarks to the Author):

In this work Goretzki, et al. studied thoroughly the role of the N-terminal intrinsically disordered region of a TRPV4 channel to understand its functional role in the context of the full length protein. This is important because most channels are rich in very long IDR that are generally not considered in structural and mechanical studies.

The conclusions of the work are that this region perform a regulatory function throughout a number of mechanisms. To reach this conclusion the authors designed and performed a number of experiments in vitro, in cell and in silico. The final picture they provide is generally consistent and robust and it is likely the most comprehensive mechanical study of a full length channel.

We thank the reviewer for this overall positive assessment.

While the work is definitely interesting and provide relevant new knowledge the paper is not always equally well crafted in its different parts and would benefit from some revisions. To me for example was very difficult to follow the accumulation of data and hypothesis tested in different directions and sometimes disconnected one from the other. One possibility could be to try to summarize all/most hypothesis at the beginning (most comes from the NMR measures) and then discussing and testing them later. It would be easy to know from the beginning what is the overall hypothesis and then see the tests, possibly in this way many section of the results could be merged and reorganised moving some data in the supporting materials and focusing more only on a key subset of them.

We have now added a paragraph to the introduction regarding the overall study motivation and re-structured some aspects of the main text to provide better readability. Specifically, we have re-arranged the sections on the role of individual IDR regions for channel activity and lipid binding. Furthermore, we have added additional technical explanations for some methods and experiments as also requested by reviewer #4.

Furthermore, the authors often used the word "integrated" to underline the fact that they used a number of techniques to investigate their system. In my opinion the word is misused because the data generated by the work are never actually integrated together in the spirit of integrative structural biology technique (cf. 10.1016/j.cell.2019.05.016), but are simply interpreted altogether.

This point by the reviewer is duly noted, and we thank them for highlighting the important primer by Rout and Sali. We have now expanded on our study's motivation and our use of the related term "integrated structural biology" in the introduction: "The lack of information on the role of additional IDR regions for channel regulation prompted us to investigate the TRPV4 IDR in more detail. In analogy to the integrative structural biology approach, which aims to build a consistent structure of a biological macromolecule or complex ((Rout & Sali, 2019, new reference 34), the central aim of our study was to derive a cohesive functional model for TRPV4 regulation by its IDR through the integration of diverse experimental and computational methods on a range of TRPV4 deletion constructs and mutants."

In our study, we used a combination of NMR and CD spectroscopy with HDX-MS to determine the degree of disorder in the TRPV4 IDR. This was then coupled to Trp fluorescence spectroscopy, cross-linking mass spectrometry and SAXS to elucidate regions of intradomain cross talk and to derive a conformational ensemble of the TRPV4 NTD as the basis for MD simulations. Our MD-generated ensembles were compared to wet-lab experimental constraints (e.g. R_g distribution, lipid interactions, etc) to elucidate the appropriate force fields. Ultimately, we used our EOM-backed and MD-generated IDR conformers to generate a structural model of the full-length TRPV4 ion channel as presented in Figure 10. This model reflects the input from various experimental sources, and notably, differs quite dramatically from an unrestrained model we proposed in an earlier publication (Goretzki et al, 2021, J. Mol. Biol. 433(17):166931) assuming worm-like behavior of the IDRs. Importantly, this new structural model now has implications for our assessment of the multifaceted regulation of TRPV4 by lipids and lipid-associated proteins. Thus, we hope that the reviewer feels that our approach does merit the label "integrated" as was also highlighted by the other reviewers.

A point-by-point review follows where I avoid discussing in-cell experiments because they do not fall in my expertise.

* Structural ensemble of the TRPV4 N-terminal intrinsically disordered region: the authors combined SEC, SEC-MALS, CD, NMR, SEC-SAXS and ensemble modelling based on SAXS for the IDR region alone and for the full N-terminal cytosolic region of the gallus gallus protein. These data indicate that the IDR is disordered, that there are transient interactions between the IDR and the folded region (e.g. for residues ~20-35 and ~55 to 115) and that, interestingly, there is also some dynamic in the folded region itself. Here my concern is about the complete lack of discussion of possible secondary structure content of the IDR region, this may be relevant to understand and model better the interaction of specific IDR elements with the remainder of the protein as well as with the membrane. The authors should try to increase the resolution of their model ideally experimentally or by simulations and/or secondary structure predictions.

We apologize that this point did not come across clearly in the paper. We have previously shown with NMR spectroscopy that the TRPV4 IDR is essentially lacking secondary structure content under different conditions (Goretzki et al, 2022, Biolmol. NMR Assign. 16(2):205-212, reference #5 in the current manuscript). We have now added an additional statement in the introduction and the result section to make the absence of appreciable secondary structure clearer, as gauged by NMR spectroscopy, supported by CD spectroscopy and apparent from our MD simulations.

* Structural dynamics of the TRPV4 ARD: Combining HDX-MS, SAXS and NMR for the folded ARD domain the authors demonstrate that this undergo a significant conformational dynamics in the peripheral ankyrin repeats. My concern here is that this is a very interesting result that is not contextualised in the overall picture. A structural model for the conformational dynamics is missing but more importantly the role of this conformational dynamics in the overall context of the paper is neither further investigated nor discussed.

We appreciate this point and the opportunity to look into the role of the ARD for TRPV4 channel regulation in more detail. To gain additional insights into the dynamics of the ARD, we have now carried

out atomistic MD simulations of the isolated ARD (new Figures 2b, c). Importantly, the simulations show high consistency with the wet lab H/D exchange experiments. As also correctly noted by the reviewer, it is very difficult and highly speculative to interpret the low resolution SAXS data for the ARD with regard to specific domain motions. Thus, we removed this part from the revised manuscript. Instead, to further contextualize a role of the ARD for TRPV4 regulation, we used MD simulations to study the dynamics of the ARD in the context of a full-length channel core embedded in the membrane (new Figures 2d, e, new supplementary movie M1, new materials and methods section "Atomistic molecular dynamics (MD) simulations"). Our results for the RMS fluctuations of the four ARDs in the channel and in isolation, in conjunction with our HDX-MS results, suggest that the ARD displays internal dynamics, but is relatively stable on the relevant timescale important for signal transduction. The ARDs are thus well suited to transmit pull forces from the IDR to the channel core. We have accordingly also extended our discussion to include this point.

* Long-range TRPV4 NTD interactions center on the PIP2-binding site: using cross-linking experiments the authors find a long range interaction between the central region of the IDR domain and the PIP2-binding site at its C-terminus, furthermore they also found inter domain interactions between the IDR and ARD domains present both when the two constructs are studied as isolated domains (Fig. 3). This finding was also verified using the intrinsic fluorescence of the solely W residue of the NTD region that is the center of the PIP2 binding site. Multiple constructs are used to show that extending the PIP2 binding site towards both the N-terminus and the C-terminus result in a decreased fluorescent emission indicative of long range interactions. This result is very elegant and robust.

We thank the reviewer for this very kind remark.

* The PIP2-binding site promotes compact NTD conformations: by mutating the PIP2 binding site from KRWRR → AAWAA and comparing the multiple constructs introduced previously the authors suggest that electrostatic is the driving force for the interaction between the PIP2 binding site and both the N-terminal part of the IDR and the ARD region. The data are convincing but they do not exclude transient interaction based on secondary structure complementarity. If electrostatic is the key would not be possible to study the unmutated constructs as a function of the ionic strength? The same observation holds for the following section ("Competing attractive and repulsive interactions between distinct IDR regions govern the NTD structural ensemble").

Comparing our CD and NMR spectroscopic data for the native IDR and IDR^{AAWAA} suggests no major differences in secondary structure content between these constructs. As pointed out above, the secondary structure content in the TRPV4 IDR is negligible, and we have not observed the formation of detectable secondary structure elements under different salt and buffer conditions (see Goretzki et al, 2022, Biomol. NMR Assign. 16(2):205-212). In the current manuscript, we show that the chemical shift differences between native IDR and its AAWAA mutant are reduced upon increasing the salt concentration from 100 to 300 mM.

To look into this in more detail, we now also compared the IDR at 100 and 300 mM NaCl using SAXS (see Figure R1).

Fig. R1: Comparison of TRPV4 IDR at 100 and 300mM NaCl using SAXS. (left) SAXS pair-distance-distribution of the IDR at low and high salt. (Center, right) The corresponding EOM analyses showing a shift in the R_g values for the IDR populations as a dependence of salt concentration in comparison to a random pool of conformers.

At the higher salt concentration, the IDR has overall larger dimensions as expressed by slight decreases of R_g and D_{max} values and according to the EOM fit, the random chain behavior of the IDR increases. These data are thus also consistent with our suggestion that electrostatic interactions mediate long range interactions in the TRPV4 IDR.

* The IDR N-terminus attenuates IDR lipid binding: the authors performed NMR, sedimentation experiments and MD simulations to investigate the interaction between the IDR and the membrane. What they found is that while the PIP2 binding site is the most important interaction site with the membrane, some interaction is also due to the central part of the sequence while the N-terminus, that was previously found to interact with the PIP2 binding site, decreases the binding interaction with the membrane likely being in competition for binding the PIP2 site. Furthermore the interaction is prevalently with negatively charged components of the membrane. The last part of the section on the possible role of the central region of the IDR domain is speculative and may be moved to discussion.

As suggested by the reviewer, we have moved the respective section to the discussion.

* A conserved patch in the IDR N-terminus mediates transient long-range interactions and autoinhibits TRPV4: here is tested a putative interaction between a conserved patch at the N-terminus and the PIP2 binding site, but all observations comes only from the observations of NMR peaks broadening in different constructs. Again the possible role of local secondary structures is not considered as well as the role of salt-concentration is not tested. This section is very speculative in comparison to other part of the paper. In the following section is then tested the possible competition between membrane and N-terminus patch in binding the PIP2 binding site in the context of channel inhibition. These two sections are a typical example of a case were the story is very difficult to follow. It would have been much easier to start from the end instead than following what may have been the consecutiveness of the experiments.

As stated in our reply to the previous comment, the lack of appreciable secondary structure content in the IDR was shown previously and salt-dependent experiments using NMR and SAXS suggest that electrostatics play an important role in intra-IDR interactions.

To improve readability, we have restructured the sections in question and accordingly also rearranged the corresponding figures in the main manuscript (Fig. 6, 7) and in the supporting information.

* Membrane-bound PIP2-binding site exerts a pull force on the ARD: here an hypothesis about a mechanical role of PIP2 binding is built using MD simulations. The authors observe that the PIP2 binding site bound to the membrane may exert a mechanical strain on the ARD domain and this may be the way this information then affect the channel. This is a very interesting hypothesis that brings the paper to conclusion and opens to something new to test, I guess that the authors should try to test this experimentally for example by inserting a very flexible sequence between the C-terminus and the PIP2 binding site of the IDR or mutating the polyproline region that may act as a spring. This may also be a point to be connected with the internal conformational dynamics of the ARD region only briefly discussed at the beginning.

We fully agree with the reviewer that this is a very interesting experiment and we are indeed planning on following up on this in future work via a library of ARD and IDR mutations. However, every new TRPV4 construct, in addition to the functional assessment, also requires a careful structural analysis to ascertain its integrity and to meaningfully tie functional and structural data together, so we hope that the reviewer agrees that this would significantly go beyond the scope of the current manuscript.

* An integrated structural model of the TRPV4 N-terminal 'belt' : this last section is a puzzle to me in the sense that there isn't any integrated structural model/ensemble produced in the paper, there is no hypothesis an no test so it is not clear which result is presented. I would either remove this part or move it to the discussion.

We apologize for any misgivings this section may have caused. The IDRs of a TRP channel have never been studied in detail, thus we feel it is important to retain this section in the manuscript. Here, we combined MD simulations with our wet lab experimental observations from SAXS, CD and NMR spectroscopy to yield an 'experimentally informed model' of the TRPV4 ion channel core in complex with its intrinsically disordered N-terminus. To make the main point of our finding clearer, we have rephrased parts of this section and updated the title of this subsection to "The cytosolic TRPV4 N-terminal 'belt' more than doubles channel dimensions beyond the structured ion channel core".

Reviewer #2 (Remarks to the Author):

The manuscript by Goretzki et al. describe a multi-faceted study of the N-terminal domain of the TRPV4 channel consisting of ankyrin repeat domain and an intrinsically disordered tail. The authors use an impressive range of experimental techniques – NMR, SAXS, hydrogen-deuterium MS, cross-linking MS as well as functional assays in a cellular system. The structural data is integrated via MD simulations. The experiments seem well-done and are analyzed appropriately. This is really as well as one can do for structural characterization of these kinds of proteins and a good example of the power of integrative structural biology. That advances significantly on the status quo and propose a plausible mechanism for how the NTD exerts its regulation of the channel via a pulling force. As such, I have no major concerns regarding publication of this manuscript.

We thank this reviewer for the very positive remarks and the appreciation of the novelty and importance of our work.

Reviewer #3 (Remarks to the Author):

The manuscript reports a structural investigation of TRPV4, which is a protein that contains both structured regions and intrinsically disordered regions (IDR). By engaging multiple structural biology tools, the authors were able to reveal the importance and the contribution of the largely overlooked intrinsically disordered domain to the function of TRPV4. Moreover, this is also a nice example demonstrating the importance of utilizing an integrated approach to solve critical structural biology questions. Regarding the mass spectrometry experiments and results of this manuscript, here are some comments that need the authors' attention.

1. The authors utilize HDX as a tool to demonstrate the structural dynamics of the protein, highlighting the highly dynamic IDR of the protein by the high deuterium uptake at 10s. Some of the regions in the IDR are not covered. The authors should comment on the low sequence coverage of the IDR in the manuscript. The low sequence coverage might be related to the stringent criteria for the peptide identification and the intrinsic limitation of online pepsin digestion, and the audience will have a better understanding of the limitations of HDX-MS experiments if the authors can call this out.

At the beginning of the respective chapter, we now added an additional section outlining the quality of the HDX-MS dataset reflected in the amino acid sequence coverage of the IDR and ARD domains and potential reasons for the low sequence coverage of the IDR..

2. In page 16, the authors mentioned that the HDX experiment was by the LEAP automation system. To the best of my understanding, the lower limit of the H/D exchange time in the LEAP system setting is 10s, but the 10s is not accounting for the mixing time that the system is taking. Upon initiating the H/D exchange, the syringe would inject the protein into deuterated buffer as described by the authors, followed by a mixing step that can take up to 60s (depending on the setting of liquid drawing speed during mixing). The H/D exchange time is counted starting from the end of the mixing step, but the actual exchange begins at the time when mixing is initiated. This delay (because of the mixing step) has a more significant impact on shorter H/D exchange time points (10s) as compared with longer points (1000s). Such a limitation needs to be verified by revisiting the experimental settings, and needs to be disclosed in the experimental section if confirmed.

We fully agree with the reviewer about the importance of the exchange times regarding the experimental set-up. In detail, the following experimental sequence was carried out: The protein was dispensed by one robot arm into a 96-well plate after which the second robot arm added the deuterated buffer. Mixing was performed (aspiration and ejection into the well again) immediately followed by aspiration of the HDX reaction and transfer to the pre-dispensed quench buffer in another 96-well plate stored at 1 °C. We checked the logs of the LEAP sample preparation system for our experiments on TRPV4 contained in the manuscript, and also confirmed the sample preparation times of the employed method manually with a stopwatch. The whole process from supplementation of D₂O buffer to the protein up to addition of the HDX reaction to the quench solution tempered at 1 °C took approximately 11-12 seconds.

We noticed that we had imprecisely stated in the material and method section that "HDX reactions were initiated by 10-fold dilution of the proteins (25 μM) in buffer [...] prepared in D₂O...", which was suggestive of the addition of the protein to a pre-dispensed D₂O buffer. Such a procedure presumably would indeed

take more time as indicated by the reviewer in the context of the LEAP system properties. However, the opposite mixing sequence was employed as laid out above; we amended the materials & methods section accordingly.

3. In page 17, authors mentioned that the cross-linked peptides were enriched by SEC before submitted to LC-MS analysis. Typically enrichment is performed before digestion, but in this case, if the primary focus is intra-molecular cross-link, this strategy may not work. However, after digestion, it might be hard to distinguish cross-linked peptides versus non-crosslinked but longer peptides. I wonder what is the principle when using SEC to enrich cross-linked species at peptide level, and how do authors make sure that this enrichment is non-discriminative. It will be nice if the authors can include more details on this important step.

Size-exclusion chromatography (SEC) was applied to enrich crosslinked peptides since crosslinked peptides are generally underrepresented in comparison to linear peptides in all crosslinking experiments. An enrichment step on the peptide level is therefore typically applied in crosslinking experiments on the peptide level (see for example recent reviews: Lee et al, Essays Biochem. 2023 (PMID: 36734207); Graziadei et al., Structure, 2022 (PMID: 34895473)). This is typically done either by cation exchange chromatography or SEC. The benefit of including a peptide SEC step directly before LC-MS/MS was first described by Leitner et al., MCP 2012 (PMID: 22286754) and leads to a relative enrichment of crosslinked peptides since they are on average larger than linear peptides (as two peptides are connected by a linker, as in the case of an intra-molecular crosslink). There is therefore no need to distinguish between cross-linked or non-crosslinked peptides at this level. An additional positive side-effect lies in the deconvolution of the spectra, which again greatly facilitates the downstream identification of the usually low-abundant cross-linked peptides. Applying SEC on the peptide level before LC-MS therefore generally increases the number of identified crosslinks.

We have now also updated the material and methods section to include the reference by Leitner et al.

Reviewer #4 (Remarks to the Author):

In this report by Goretzki and colleagues, the authors use an ensemble of approaches to characterize the role of the intrinsically disordered N-terminal tail of TRPV4. Previous studies identified two regulatory motifs in the N-terminal domain, a proline-rich domain and a PIP2-binding site. Here, the authors identify an autoinhibitory patch in the N-terminal domain that can compete with PIP2 for the PIP2 binding site and that disruption of the interaction between the PIP2 binding site and the membrane lessens the force exerted on the structured domains of TRPV4 by the N-terminal domain. Based on these analyses, the authors propose a hierarchical model for disparate stimuli can regulate the activity of TRPV4. While these results are interesting and can potentially improve the understanding of this channel, the model is inadequately validated, and direct functional assessments of the model should be performed prior to publication.

Major comments:

1. As part of this work, the authors use a wide array of approaches. However, a more complete description of the various approaches and especially a description of their limitations would be helpful for the broad readership of Nature Communications.

We apologize for this inconvenience. The length of a Nature Communications manuscript has been prohibitive to explain the methods in great detail. We have now expanded upon our study motivation in the introduction and added additional remarks on the scope of methods used at individual sections where we felt readers would most likely benefit. We have also restructured some sections to improve readability as also requested by reviewer #1.

2. The Ca²⁺ imaging data is difficult to interpret due to the large differences in the expression of mutant channels as shown in Figure S7. How was the difference in expression controlled for in these assays?

The reviewer is correct that there are differences in expression levels of the different TRPV4 constructs as indicated by our Western Blots. However, the lowest expressing construct (TRPV4 Δ N68) has the highest, while the constructs with the highest expression (TRPV4 Δ N111, Δ N118, Δ N133) display the lowest calcium levels in response to hypotonic stress. The effects we describe in the manuscript are thus rather underestimating the consequences of the deletions, therefore we do not deem the differences in expression problematic for the conclusion we are drawing.

3. There also appears to be several GFP-positive fragments in each lane. Do the lower bands correspond to degradation products and how would degradation influence the activity of the channel?

The additional bands appear to be non-specific rather than degradation products (see Fig. R2).

Fig. R2: Uncropped Western Blot of TRPV4 expression in MN-1 cells. On the left is the image shown in Fig. S7a. On the right, the same blot with longer exposure times is depicted.

4. The authors propose that the NTD regulates TRPV4 through a hierarchical network of lipid-dependent interactions. However, this model is not directly tested in manuscript. Electrophysiological recordings of excised patches from cells expressing wild-type and mutant channels in the presence and absence of PIP₂ and other anionic lipids is necessarily to demonstrate the validity of the proposed model.

The reviewer is correct that we provide a structural model how lipids modulate TRPV4 function via IDR interactions. This was exactly the aim of this paper, as the importance of the so-called PIP₂-binding site for lipid-dependent TRPV4 function has been demonstrated by different labs previously (e.g. Garcia-Elias et al., 2013, PNAS; Caires et al., 2022, Cell Reports). Our study now provides a molecular mechanism for these observations through our findings that newly discovered regulatory elements of the TRPV4 IDR directly interact with this basic stretch (e.g. Fig. 6), that the entire IDR interacts with lipids (e.g. Fig. 7, Fig. 8) and that the interaction between remote IDR regions competes with PIP₂ binding (Fig. 8). Importantly, the details of intramolecular IDR or IDR-lipid interactions at a single amino-acid resolution have never been looked at for any TRP channel and became only possible to elucidate with the combination of techniques used here. Together with cell based functional assays, this structural approach was then extremely powerful to demonstrate that it is the newly discovered interplay of strictly lipid-dependent interactions within the IDR that determine channel responses to osmotic stimuli.

5. Throughout the manuscript the authors employ deletion constructs to analyze the role of specific domains. However, by deleting large regions of the protein, they exclude the possibility that non-specific interactions can contribute to the regulation of the channel. The authors should repeat some of the experiments with constructs in which the sequences of the regions being tested are scrambled rather than deleted to allow discrimination between specific and non-specific effects.

The reviewer would be correct if we were dealing with defined binding sites between different IDR regions. However, the transient nature of the observed contacts, the fluidity of the contact sites and the fact that these interactions depend on electrostatic complementarity, all severely complicate the classic scrambling approach.

Indeed, we would go as far as to say that with minor exceptions, such as the interaction between regulatory Patch and PIP₂ binding site, which we investigated extensively both in the native, full-length NTD by non-invasive methods such as NMR and Trp fluorescence spectroscopy and then verified through additional, targeted mutagenesis experiments to complement our deletion mutants, it is exactly the sum of a multitude of weak and mostly unspecific intra-IDR and IDR-membrane interactions that “shape” the TRPV4 NTD structural ensemble. It can be speculated that this endows the channel with sufficient flexibility to respond to a diversity of stimuli and partners.

Reviewer #5 (Remarks to the Author):

Goretski et al.

Crosstalk between regulatory elements in the disordered TRPV4 N-terminus modulates lipid-dependent channel activity

The manuscript submitted by Goretski et al. aims to characterize the regulatory elements within the TRPV4 IDR completely. Regulation of many classes of proteins (ion channels included) by attached or interacting IDRs is critical to many functions. Despite this, only a handful of frequently occurring IDR regulatory motifs are well understood. I am impressed by the work done in this manuscript to understand the structure/disorder function relationship in the TRPV4 ion channel. I would go as far as to say that this manuscript puts on display one of the most rigorous structural analyses of an IDR I have encountered. The authors deploy nearly all of the canonical experimental techniques used to study IDRs (NMR, SAXS, HDX, SEC-MALS, CD spec, simulation) and combine these with functional assays in a cellular background. I have no hesitation in recommending this manuscript. The only additional experiments I would consider suggesting would be far outside the scope and only occur to me due to how complete the manuscript already is (FRET experiments in the full-length constructs in a membrane environment?).

We thank the reviewer for the very positive response to our manuscript. Of course, we also dream of a dynamic picture of the full-length channel in (native) membranes, but as the reviewer rightfully remarks themselves, this would be far beyond the scope of the current manuscript.

I have a few suggestions that may improve the final product.

The manuscript does display a dizzying array of techniques. I believe it would help the reader if the experimental plan and the significant conclusions were clearly on display in the introduction.

We have now expanded on our motivation in the introduction, and, as also requested by reviewers 1 and 3 at different points in the manuscript where we felt additional descriptions would be beneficial to the reader.

Some analyses are a bit redundant. There is little need to show the Pr distribution, and the EOM fits. They provide more or less the same information, given that the underlying conformations in the EOM ensembles are not interpretable (not that there is an attempt to interpret them here). I would forgo either example in favor of analysis with CG or atomistic simulations, but that is nitpicking.

We thank the reviewer for this important comment. The reason we chose to present the analyses as shown was that the $p(r)$ comparisons provide a very straightforward visual aid demonstrating very different structural regions within the NTD. In particular, we consider that the $p(r)$ comparisons nicely show that the NTD encompasses two disparate regions consisting of the more compact ARD connected to the more expansive/structurally heterogeneous IDR, and that the IDR itself has a skewed distribution of states that, as a population, tends toward anisotropic sampling. The comparative R_g distribution results from EOM – for both NTD and the IDR – go to qualify the relationship internal to, and between, the ARD and IDR of the NTD that the $p(r)$ profiles in-and-of themselves do not show. For example, compared to a randomly generated pool of structures, both the NTD and IDR ensembles have sub-populations that can indeed sample extended/expanded states, however the R_g maximum frequency of the refined pools for both proteins are less than what is expected for a purely random set of structures indicating that transient intrachain interactions are present within the IDR, demonstrating that sub-populations of structures exist within the total ensemble that exhibit chain collapse. Indeed, it appears that these types of sub-populations are the more 'preferred resting state', compared to the less frequent expanded/extended states within the IDR or NTD populations.

Therefore, we believe that by displaying both the $p(r)$ and the EOM results together, the reader obtains a sense of both the 'overall/global view' of structural state sampling, i.e., from the $p(r)$, combined with a more 'refined' view of the of sub-population occupancies within the ensembles, i.e., from EOM. We would thus kindly ask to keep the respective figures in their current form.

To make this point clear for the reader, we have now amended figure legend Fig 1f to state 'Every protein exhibits levels of conformational heterogeneity and the $p(r)$ profiles should be interpreted as the summed volume-fraction weighted contribution within the sample population, and not as single-particle distributions.'

Similarly, I think that panels b-d in figure one are a bit redundant. I believe information about the quality of the protein can be moved to the supplemental information.

We appreciate the comment by the reviewer but feel that it is a rather unfortunate development in recent years that the quality of the protein preparation is no longer prominently displayed in manuscripts but rather hidden in the supplementary information. The protein preparation is, after all, the foundation of all other experiments and analyses that follow. In particular for proteins with large intrinsically disordered regions such as TRPV4, clean sample preparation is not trivial. Furthermore, for the initiated reader, these data already hold important clues. For instance, the SDS-PAGE shows the expected behavior with all proteins migrating according to their respective size, while our SEC-MALS data already alludes to the complex structural nature of the TRPV4 N-terminal domains. We thus would politely request that the data be shown as they currently are.

The purples and greys in figure 1 are a bit hard to distinguish (particularly when printed)

We sincerely apologize for this. We have tried numerous combinations of colors and due to the large number of constructs used in this study throughout all figures (including SI), feel we have run out of viable alternative options.

The classifications of protein flexibility used in fig S1d are unnecessarily fine-grained and muddy the interpretation. The definitions of Uversky are up for debate. It would be more clear to simply label known reference states such as "folded," "theta-chain," etc. The SARW reference state is already used as a point of comparison for the SAXS data.

We thank the reviewer for this important point. The aim here was to analyze the quality of the protein preparation and to assess possible deviations from fully folded or random coil states. We have now rephrased the legend for Fig. S1d to reflect this ("The ARD can be generally classified as more-compact native-like protein, the IDR as a native coil-like protein and the NTD as a pre-molten globule-like protein, i.e. a mixture of unfolded and globular states.")

The fact that a D_{max} of ~ 11.5 nm is required to fit data from a globular protein indicates that larger structures are present in the data. I would also say that in the norm Kratky plot (S3e), the ARD might not look like a pure globule. To my eyes, the peak is located at $qR_g > \sqrt{3}$. I suspect that adding the reference point ($\sqrt{3}$, 1.1) to the Kratky plot would further support your conclusions.

The reviewer is correct, the peak maximum is at $qR_g > \sqrt{3}$. We have now updated figure S3e and its legend to account for this ("The maximum at $qR_g = \sqrt{3}$ and $(qR_g)I(q)I(0)^{-1} = 1.1$ representative of an ideally globular protein is indicated.")

REVIEWERS' COMMENTS

Reviewer #1 (Remarks to the Author):

The authors have improved the manuscript and addressed all relevant concerns. This work is remarkable for the amount of data produced and it provide a quite comprehensive picture of the role of disordered regions in the full length TRPV4.

Reviewer #4 (Remarks to the Author):

The authors have substantially revised and improved the manuscript. If the authors do wish to perform electrophysiological analyses of the effects of PIP2 on wild-type and mutant channels, a more thorough discussion how the existing electrophysiological analyses complements their work would be critical for interpreting their studies. With this change, the manuscript would be suitable for publication in Nature Communications.

Reply to the Reviewers

Reviewer #1 (Remarks to the Author):

The authors have improved the manuscript and addressed all relevant concerns. This work is remarkable for the amount of data produced and it provide a quite comprehensive picture of the role of disordered regions in the full length TRPV4.

We thank the reviewer one more for their important insights in the first round of revisions which significantly improved the paper and the kind words they find for our work here.

Reviewer #4 (Remarks to the Author):

The authors have substantially revised and improved the manuscript. If the authors do wish to perform electrophysiological analyses of the effects of PIP2 on wild-type and mutant channels, a more thorough discussion how the existing electrophysiological analyses complements their work would be critical for interpreting their studies. With this change, the manuscript would be suitable for publication in Nature Communications.

We thank the reviewer for the time taken to review our manuscript once more and their acknowledgement of the additional work included after the first round of revisions. Currently, we are not planning electrophysiology analyses, as the questions addressed in our paper, the biophysical characterization of the TRPV4 IDR and the analysis of lipid dependent crosstalk on an atomic level, cannot easily be looked at by this method. There are currently no electrophysiology studies available on TRPV4 (or another TRP channel) that investigate a hierarchy of regulatory elements in a cytosolic TRP channel IDR, as this concept was newly established by us in the current manuscript. Other researchers that looked at TRPV4 activity commonly used Calcium imaging as a well-established and well-suited method, including our own previous work (see e.g. McCray et al, 2021, Nature Commun). We thus hope that the reviewer can agree with us that the combination of numerous complementary functional and biophysical methods in our manuscript are sufficient to support our finding that the TRPV4 IDR can be segregated into a hierarchy of regulatory motifs and harbours a master autoinhibitory element in its N-terminal half.